# Frequent pauses in *Escherichia coli* flagella elongation revealed by single cell real-time fluorescence imaging

Ziyi Zhao [1], Yifan Zhao[1], Xiang-Yu Zhuang[2], Wei-Chang Lo[3], Matthew A.B. Baker[4], Chien-Jung Lo [2] & Fan Bai[1]

The bacterial flagellum is a large extracellular protein organelle that extrudes from the cell surface. The flagellar filament is assembled from tens of thousands of flagellin subunits that are exported through the flagellar type III secretion system. Here, we measure the growth of *Escherichia coli* flagella in real time and find that, although the growth rate displays large variations at similar lengths, it decays on average as flagella lengthen. By tracking single flagella, we show that the large variations in growth rate occur as a result of frequent pauses. Furthermore, different flagella on the same cell show variable growth rates with correlation. Our observations are consistent with an injection-diffusion model, and we propose that an insufficient cytoplasmic flagellin supply is responsible for the pauses in flagellar growth in *E. coli*.

[1] Biodynamic Optical Imaging Center (BIOPIC), School of Life Sciences, Peking University, Beijing 100871, China. [2] Department of Physics and Graduate Institute of Biophysics, National Central University, Jhongli, Taiwan 32001, Republic of China. [3] Department of Physics, Duke University, Durham, NC 27708, USA. [4] School of Biotechnology and Biomolecular Science, University of New South Wales, Sydney, Australia. These authors contributed equally: Ziyi Zhao, Yifan Zhao, Xiang-Yu Zhuang. Correspondence and requests for materials should be addressed to C.-J.L. (email: cjlo@phy.ncu.edu.tw) or to F.B. (email: fbai@pku.edu.cn)

Flagellated bacteria swim using the bacterial flagellar motor, which consists of a reversible rotary motor, a short proximal hook and a long helical filament[1–5]. Using the transmembrane electrochemical ion motive force to power the bacterial flagellar motor[6–11], fast rotating flagella can propel the cell body at a speed of 15–100 μm/s. Previous research has established that a flagellum is assembled from the inside out, beginning with the basal body embedded in the cell membrane, and that the structural components of the sequential rod, hook, and filament are unfolded and exported through a type III flagellar secretion system (T3FSS)[12–15].

Details of the molecular mechanism underlying flagellar assembly remain poorly understood. Earlier studies using electron microscopy[16] or dark-field microscopy[17] suggested that the flagellar growth rate decreased exponentially with length. By contrast, Turner et al.[18] measured growing segments that were sequentially labeled with different fluorophores and found the average flagellar growth rate of *Escherichia coli* to be independent of length, showing a relatively constant growth speed. More support for a model predicting constant growth was given by Evans et al.[19], who showed evidence of successive head-to-tail linkages between unfolded flagellin subunits in the transit channel. However, these measurements were all based on population measurements of flagellar lengths[20].

Recently, using single-cell real-time imaging in *Vibrio alginolyticus*, Chen et al.[21] revealed that the growth rate of each single polar flagellum first plateaued and then decayed with length. At the same time, Renault et al.[22] demonstrated for peritrichous *Salmonella enterica* that the growth rate of a single flagellum was inversely correlated with flagellar length. These single-cell experiments provided insight into the molecular mechanism of flagellar assembly and confirmed that flagellar growth rate decays with filament length in two species.

In this paper, we return to focus on the growth rate of *E. coli* and determine whether the constant flagellar growth observed in *E. coli* is the exception[18, 23]. We first demonstrate an efficient imaging strategy to observe and measure the flagellar growth of *E. coli* with high spatial and temporal resolution. By engineering flagellins with an optimized tetracysteine tag (TC tag) and labeling them with biarsenical dyes[24, 25], we performed real-time and dynamic single-flagellar growth tracking. In addition, since *E. coli* is peritrichous, we also investigated the extent of correlation in growth between multiple flagella on separate filaments. Our results revealed several distinct features in the flagellar growth of *E. coli*, and we provide a mathematical framework that explains the physical principles underlying bacterial flagellar assembly.

## Results

**Real-time imaging of flagellar growth by FlAsH/ReAsH labeling**. We chose the tetracysteine tag/biarsenical dyes system[24, 25] to label and measure the flagellar growth of *E. coli*. Because of the small size of the tetracysteine tag, bacterial motility and flagellar growth were not perturbed[25, 26] (Supplementary Note 1 and 2, Supplementary Figs. 1 and 2). As a membrane-permeant biarsenical dye[24, 27], FlAsH/ReAsH enables live cell imaging without fixing the cell. To further reduce the background fluorescence and increase image quality, we improved the labeling method[25] by inserting an optimized tetracysteine tag (FLNCCPGCCMEP, TC tag)[28] into bacterial flagellin (RP437-TC-FliC, Fig. 1a), which increased the binding affinity of biarsenical dyes to the tetracysteine motif. Consequently, FlAsH/ReAsH remained non-fluorescent until it is bound to the tetracysteine tag with high affinity and specificity, becoming strongly green/red fluorescent. Therefore, high-resolution fluorescence images of flagella with good contrast could be obtained without washing away the labeling reagent (Fig. 1b), allowing the subsequent staining of newly grown segments and the real-time imaging of flagellar growth.

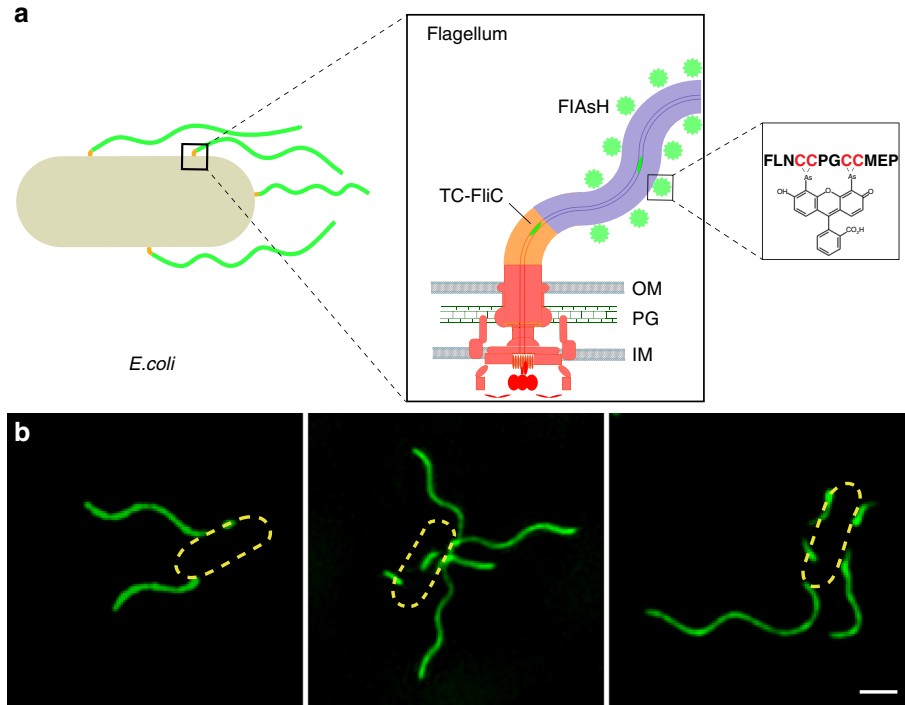

**Fig. 1** Bacterial flagella of *E. coli*. **a** A schematic illustration of the optimized tetracysteine tag (FLNCCPGCCMEP) inserted flagellins labeled by the biarsenical fluorophore FlAsH. **b** Representative super-resolution fluorescent images of *E. coli* peritrichous flagella acquired by the Structured Illumination Microscopy (SIM). Scale bar =1 μm

**Growth rate of *E. coli* flagella decays on average as they lengthen**. We first examined *E. coli* flagellar growth in a large population of cells. To improve the accuracy of flagellar length measurements, super-resolution fluorescence images were captured using a Nikon Structured Illumination Microscope (N-SIM), in which one pixel corresponded to a physical distance of 30 nm. Flagella were labeled with FlAsH/ReAsH successively two or three times (double-color and triple-color labeling, Fig. 2a) with a time interval of 30 min (Methods)[22], which provided a higher temporal resolution than the previously reported interval of 3 h[18]. In Fig. 2b, we plotted the relationship between the new extension length of the filament (secondary fragment) and its initial length (primary fragment) of 311 flagella. From this data we observed that the flagellar growth rate of *E. coli* exhibited large variations at similar initial lengths. As shown in Fig. 2c, the secondary fragments, which grew from basal filaments, could elongate variably (0–1400 nm) during a 30 min interval, which was consistent with a previous report[18]. However, the average length of the secondary fragment decreased from ~0.8 µm to ~0 µm as the primary fragments became longer (Fig. 2c). Similar results were achieved by measuring the flagellar growth of *E. coli* cells attached to the coverslip using spinning disk confocal microscopy (Supplementary Note 3 and Supplementary Fig. 3) and the washing steps during sample preparation did not affect the physiology of the cells (Supplementary Note 5 and Supplementary Fig. 5c).

Next, triple-color labeling of filaments was used to measure growth rate at three separate intervals. In Fig. 2d, 157 flagella were binned into 5 groups according to their primary segment lengths and the distributions of secondary versus tertiary segment lengths of each group were plotted. Consistent with our double-color

labeling results, the extension length of the tertiary fragment inversely correlated with its initial length (Fig. 2e, f). Altogether, double-color and triple-color labeling experiments demonstrated that flagellar growth of *E. coli* showed large variations but on average decayed with respect to flagellar length.

**Single flagellar growth tracking identified frequent pauses**. To investigate why the flagellar growth rates vary significantly among filaments of similar length, we tracked single flagellar growth in real-time. Figure 3a, b gives two representative examples showing the dynamic process of single-flagellar growth recorded by time-lapse imaging. In Fig. 3c, the corresponding flagellar length vs. time curve is presented and the instantaneous flagellar growth rate was simply calculated as the length difference between two adjacent images divided by the time-lapse interval of 10 min. Previous single-cell studies concluded that flagellar growth of both *Vibrio alginolyticus* and *Salmonella enterica* decreases with length[21, 22]. Interestingly, our data indicated that a single flagellum of *E. coli* grew with frequent pauses over its entire growth period. We found that clear increases and decreases in growth rate occurred during flagellar elongation (Fig. 3c). An additional 16 single-flagellar growth traces were collected (Fig. 3d) and fluctuation in flagellar growth rates was widely observed in *E. coli*. To quantify the pausing frequency during single-flagellar growth, we defined a pause event as when the growth rate was lower than 5 nm/min (Methods) between two successive frames in a 10-min time interval. We also examined the distribution of pause events with respect to their flagellar growth across all traces (Fig. 3e) and found that pauses in growth can occur at different flagellar lengths. To further prove

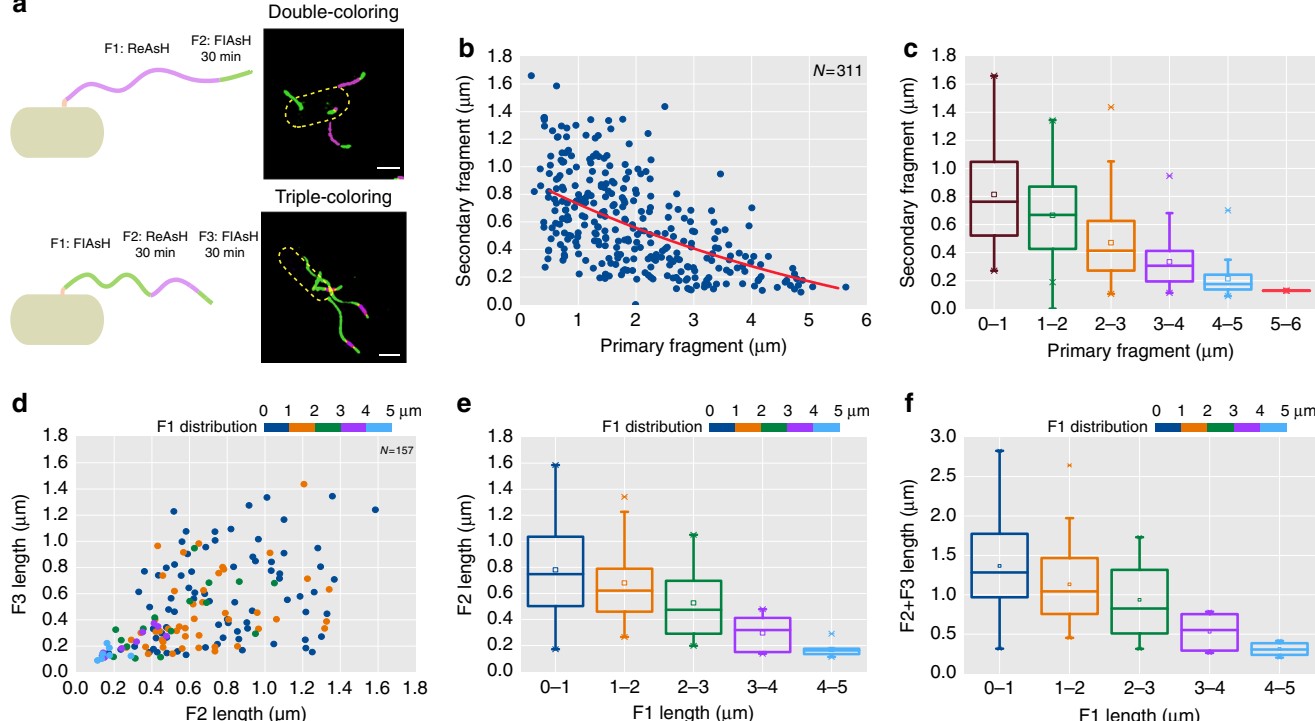

**Fig. 2** Flagellar growth rate measurements using super-resolution imaging. **a** Examples of TC tag inserted flagella labeled by double- and triple-color labeling. The secondary (F2) and tertiary (F3) segments were grown for 30 min, respectively. Scale bar =1 µm. **b** Plot of lengths of secondary versus primary segments (F1) for double-color labeling, fitted by an exponential line with mean values taken from **c**. A total of 311 flagella were measured and analyzed. **c** Box plot of the data from **b** in which 1 µm was set as the bin size value, showing that flagellar growth rate decreases with increasing flagellar length (center line, median; center square, mean; box limits, upper and lower quartiles; whiskers, ×1.5 interquartile range). **d** Relation between the lengths of secondary and tertiary segments. Overall, 157 individual flagella were measured and divided into 5 groups, as indicated by colors according to the lengths of primary segments. **e, f** Box plots of secondary and secondary plus tertiary segments according to the length of F1 from **d**, respectively

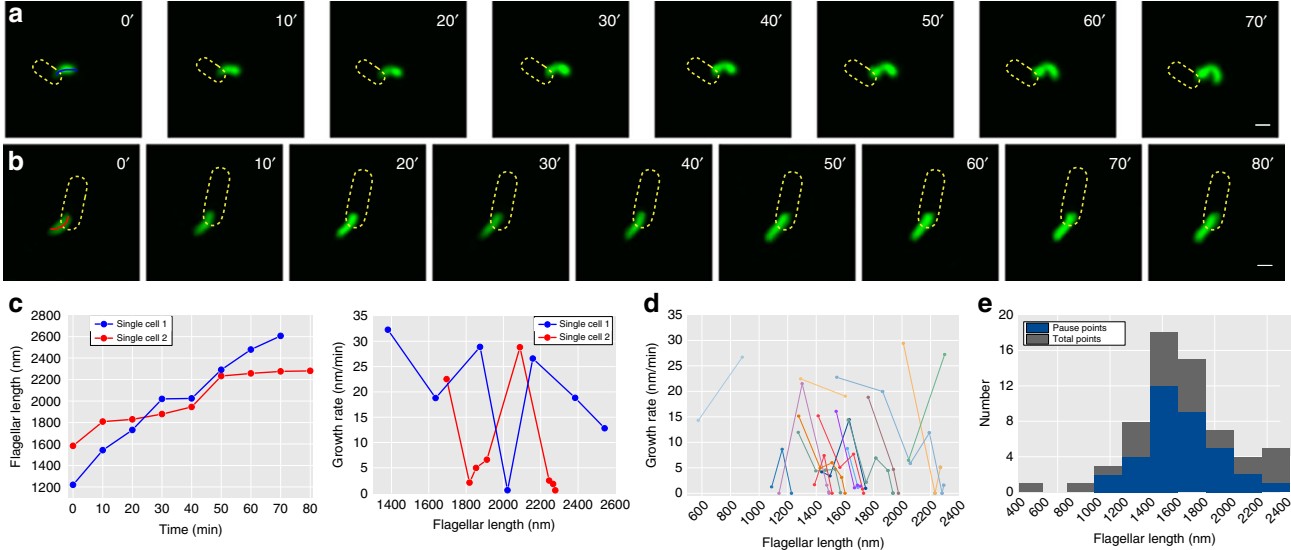

**Fig. 3** Real-time time-lapse fluorescence imaging of single flagellar growth. **a**, **b** Continuous recording of a single growing flagellum with images taken every 10 min. Other flagella on the cell were removed to show the growing flagellum more clearly. Scale bar = 1 μm. **c** Plots of flagellar length versus time and the flagellar growth rates versus flagellar lengths (Blue line: the flagellum in **a**; Red line: the flagellum in **b**). **d** More growth measurements of 16 individual flagella tracked by real-time imaging. **e** Distribution of pause events (Blue) occurring in the total collected data points (Dark gray) at different flagellar lengths, binned at 200-nm intervals. All data points are from the measurements in **d**

that the observed pauses in flagellar growth were not an artifact caused by surface coating with poly-L-lysine or different medium, we repeated the experiment with cells immobilized on an agarose pad while the flagella were free to move in space and estimated the amount of flagellin in rich and minimal medium. Similar results of cell physiology (Supplementary Note 6, Supplementary Figs. 5a, b and 7) and flagellar growth pattern (Supplementary Note 4 and Supplementary Fig. 4) were obtained, indicating that pausing is an intrinsic feature of *E. coli* flagellar growth.

**Growth rates of different flagella on the same cell**. We next studied the growth rates of different flagella on the same cell. The approach of double-color and triple-color fluorescence imaging at a 30-min interval (as described in Fig. 2a) was used again to measure lengths across a large population. Figure 4a provides six representative super-resolution images showing the double-color labeling of multiple flagella. The relationship between secondary fragments (green) and primary fragments (purple) were plotted and the data points from each same cell were connected. For triple-color labeling, two successive newly born segments of different flagella grown on the same cell were analyzed (Fig. 4b). We analyzed the growth of the secondary fragments (purple) with respect to the primary fragments (green) and also the tertiary fragments (green) were compared with the total growth from the secondary and primary fragments (purple and green). Interestingly, in these single filament examples, the extension lengths of multiple growing flagella on the same cell were all located below the black curve, indicating that the growth rates of multiple filaments on the same cell were uniformly slower than the average growth speed of a large population. These results imply that the growth rates of different flagella on the same cell are likely to be influenced by a global factor and thus showed a correlation.

The growth of multiple flagella on the same cell was also monitored by real-time fluorescence recordings. Figure 4c, d shows two representative examples of growth tracking images of multiple flagella growing on one cell. The corresponding flagellar length vs. time curves are presented and the instantaneous flagellar growth rate is calculated as the length difference between

two adjacent images divided by the time interval of 10 min. The growth rate measurements of single flagella are consistent with above results that the flagella of *E. coli* grow in a punctuated manner with frequent pausing. In addition, we observed that elongation of different flagella on the same cell showed similar pace, suggesting that peritrichous flagellar growth might be intrinsically correlated at a cellular level. However, it was difficult to follow the growth of multiple flagella and calculate their cross-correlation over long periods, because flagella often become entangled as they lengthen.

**Dynamic flagellin concentration causes pauses in flagellar growth**. We next sought to understand what causes the frequent pauses in *E. coli* flagellar growth. According to previous theoretical works[21–23, 29], the flagellar growth rate is primarily determined by the diffusion coefficient of flagellins in the channel and the loading speed of new flagellins into the channel by the secretion apparatus, which in turn depends on the cytoplasmic flagellin concentration. Therefore, we postulated that the fluctuation in intracellular flagellin concentration contributes to the fluctuations in *E. coli* flagellar growth rates.

To test this hypothesis, we constructed a flagellin over-expression assay (Methods). In the host strain RP437-TC-FliC, plasmid pBAD18-TC-FliC could be induced by arabinose to overexpress TC motif tagged FliC (Fig. 5a). Using western blotting, we confirmed that the total amount of flagellin sufficiently increased under 0.2% arabinose induction (Fig. 5b, Supplementary Note 7 and Supplementary Fig. 8), and that cell growth was nearly identical with or without arabinose induction (Supplementary Fig. 6).

Double-color SIM imaging was again applied to quantify the effect of overexpressed flagellins on flagellar growth during the same time period and under identical experimental conditions. The length comparisons of secondary versus primary segments of arabinose-induced strain (blue) with uninduced strain (gray) are shown in Fig. 5c. The average growth value of secondary segments in a 30-min time interval is also calculated (Fig. 5d). Flagellar growth of the strain induced with arabinose was faster on average,

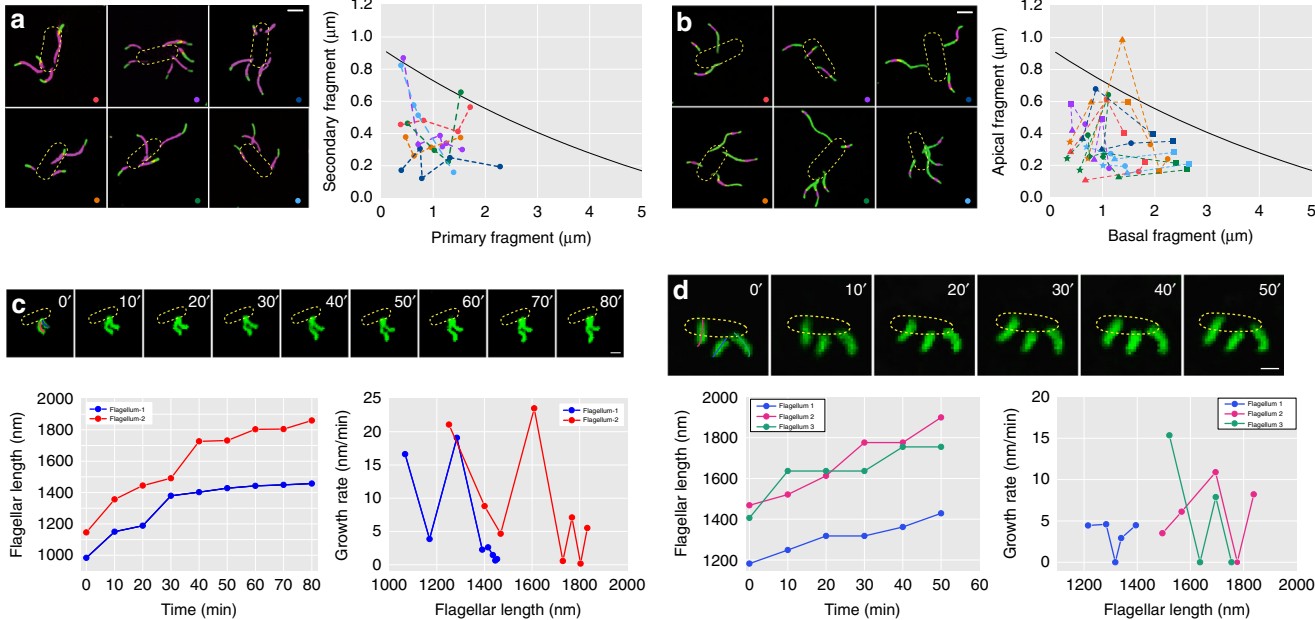

**Fig. 4** Growth measurements of multiple flagella on the single cell. **a** Double-color and **b** triple-color labeling were used to record flagellar growth of several *E. coli* cells. For the double-color labeling in **a**, the relationship between secondary (green) and primary fragments (purple) are plotted, and the data points from the same cell are connected. For the triple-color labeling shown in **b**, the secondary (purple) versus primary fragment (green) lengths and the tertiary (green) versus secondary plus primary fragment (purple and green) lengths were analyzed. The data points corresponding to each individual flagellum are indicated with the same symbol and flagella from each individual cell are in the same color. The exponential fit to the mean growth value vs. flagellar length relationship from Fig. 2c is overlaid in black. Scale bars = 1 μm. **c, d** Real-time fluorescence tracking was performed to observe different flagella growing on a single cell. Plots of flagellar length versus time and the growth rates versus flagellar lengths are presented

implying that the pauses that we observed could be partially recovered by increasing the cytoplasmic flagellin concentration.

We then repeated single flagellar growth tracking to compare individual flagellar growth under different flagellin expression levels. We quantified the change in pausing frequency, defined as the ratio of pause events over total recording (Fig. 5e). Under arabinose induction, the pausing frequency of the flagellin overexpression strain was ~1/2 of that of the RP437-TC-FliC. Therefore, overexpression of flagellin proteins significantly reduced the number of pauses in *E. coli* flagellar elongation, thus confirming that an insuffcient flagellin supply is a major factor in pausing.

**Simulating flagellar growth by an injection-diffusion model.** The injection-diffusion model is currently the leading model for the bacterial flagellar secretion system[21–23]. In the model, the flagellar secretion apparatus injects flagellin monomers into the central channel and then flagellins diffuse along the channel until they arrive at the distal end where they polymerize to form a new flagellin extension.

This is illustrated as a schematic in Fig. 6a. (1) The Brownian motion of each individual flagellin in the channel can be simulated by the Langevin equation:

$$m_i \frac{d^2 x_i}{dt^2} = F_{\text{push}} + F_{\text{drag}} + F_{\text{Brownian}} + F_{\text{repel}} \quad (1)$$

where $m_i$ and $x_i$ denote the mass and position of the $i$th flagellin monomer in the channel, respectively.

(2) There are four different forces that can act on the flagellin during this self-assembly process. The T3FSS utilizes free energy derived from ATP hydrolysis and proton motive force (PMF)[11] to inject flagellins into the channel. Therefore, we assume that flagellins at the entrance of the channel are pushed by a constant force $F_{\text{push}}$ until a whole flagellin subunit is inserted into the

channel. In our simulation,

$$F_{\text{push}} = F_0 \quad (2)$$

for the flagellin being pushed at the entrance, and

$$F_{\text{push}} = 0 \quad (3)$$

for flagellins in the channel.

(3) $F_{\text{drag}}$ represents the viscous force and is defined as,

$$F_{\text{drag}} = -\zeta \frac{dx_i}{dt} \quad (4)$$

where $\zeta$ is the viscous drag coefficient. $F_{\text{Brownian}}$ simulates the Brownian force due to thermal fluctuations,

$$F_{\text{Brownian}} = \eta(t) \quad (5)$$

where $\eta(t)$ satisfies the white-noise property,

$$\langle \eta(t) \rangle = 0 \quad (6)$$

$$\text{cov}[\eta(t)\eta(s)] = \langle \eta(t)\eta(s) \rangle = 2kT\zeta\delta(t-s) \quad (7)$$

(4) When a flagellin diffuses into the channel, it is partially unfolded as a peptide chain which are ~1 nm in diameter by 74 nm long ($L_{\text{FliC}}$)[23, 30]. To simulate the interaction between adjacent flagellins, we modeled partially unfolded flagellin as a one dimensional particle of length $L_{\text{FliC}}$. The repulsion force between any pair of flagellins can be simulated by the Lennard–Jones potential,

$$F_{\text{repel}} = 4\varepsilon \sum_{j \neq i}^{n} \left( \frac{12\sigma^{12}}{x_{ij}^{13}} - \frac{6\sigma^6}{x_{ij}^7} \right) \vec{x_{ij}} \quad (8)$$

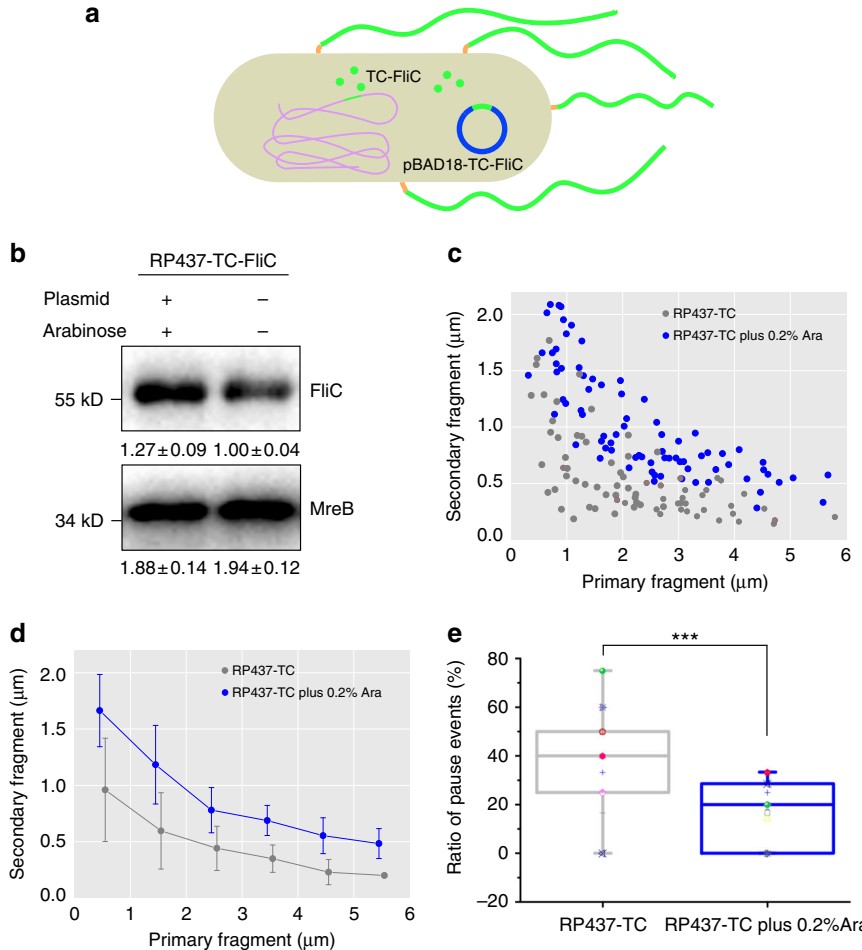

**Fig. 5** Pauses in flagellar growth can be partially recovered by overexpression of FliC. **a** Schematic of flagellin overexpression strategy for the RP437-TC-FliC strain carrying plasmid pBAD18-TC-FliC. **b** Top: Immunoblotting of total flagellin (FliC) of RP437-TC-FliC strains with or without expression of a plasmid-borne RP437-TC-FliC gene. Bottom: Detection of cytoskeleton protein MreB was used as a loading control. The bands were quantitated by ImageJ (relative flagellin levels report Mean ± S.D., $n = 3$). Mann-Whitney test was performed defining differences as significant, **$P < 0.05$. Full blots are shown in Supplementary Fig. 9a. **c** Scatter plot of the lengths of secondary versus the primary segments from which they grew for 30 min by SIM double-color imaging (For two cultures, RP437-TC-FliC strains with or without plasmid expression, 75 and 77 flagella were analyzed respectively.). **d** Average length of the secondary segments during 30 min for the two cultures in **c**, binned at a 1-μm interval. Error bars are standard deviation. **e** The comparison of flagellar growth between two cultures (RP437-TC-FliC strains with or without plasmid expression) performed by real time imaging of single flagellar growth. Flagellar growth rate slower than 5 nm/min between two frames in a 10-min interval is designated one pause event. ***$P < 0.01$. The number of flagella analyzed per group was 13 and 14, respectively

where ε, σ, and $\vec{x_{ij}}$ are the L–J potential depth, radius, and unit vector, respectively. Finally, when an unfolded flagellin travels to the distal end of the flagellum, it crystallizes immediately and extends the flagellum by $\Delta L$[23, 30].

(5) Since the environment where protein stays is one of low Reynolds number, inertial forces can be neglected and removed from the Langevin equation[31, 32]. Thus, Equation 1 can be simplified as

$$\varsigma \frac{dx_i}{dt} = \eta(t) + F_{push} + F_{repel} \quad (9)$$

By introducing a Wiener process, Equation 9 can be numerically simulated:

$$x_i(t + \Delta t) = x_i(t) + \frac{1}{\varsigma} \left[ \sqrt{2D\Delta t} \cdot W + F_{secret}\Delta t + F_{repel}\Delta t \right] \quad (10)$$

where $W \sim N(0, 1)$ Gaussian white noise, $D$ is the apparent diffusion coefficient of a flagellin monomer in the channel, and $\Delta t$ is the time interval of one simulation step.

(6) At the base of the channel, we assume that the export apparatus loads new flagellin into the channel at rate $R$, which is proportional to FliC concentration in the cytoplasm. To explain the high variation of flagellar growth rates (Fig. 2b), punctuated flagellar growth rate (Fig. 3) and high growth rate correlation within the same cell, we speculate that there is a dynamic change in the cytoplasmic FliC concentration causing the loading rate variation. For simplicity, we tested a sinusoidal function to mimic periodic flagellin loading rate (Fig. 6b),

$$R(t) = \overline{R} + R_0 \sin\left(\frac{2\pi t}{T_s} + \varphi\right) \quad (11)$$

where $\overline{R}$, $R_0$, $T_S$, and $\varphi$ are the basal secretion rate, secretion variation strength, secretion variation period, and secretion variation phase, respectively. More computational details and

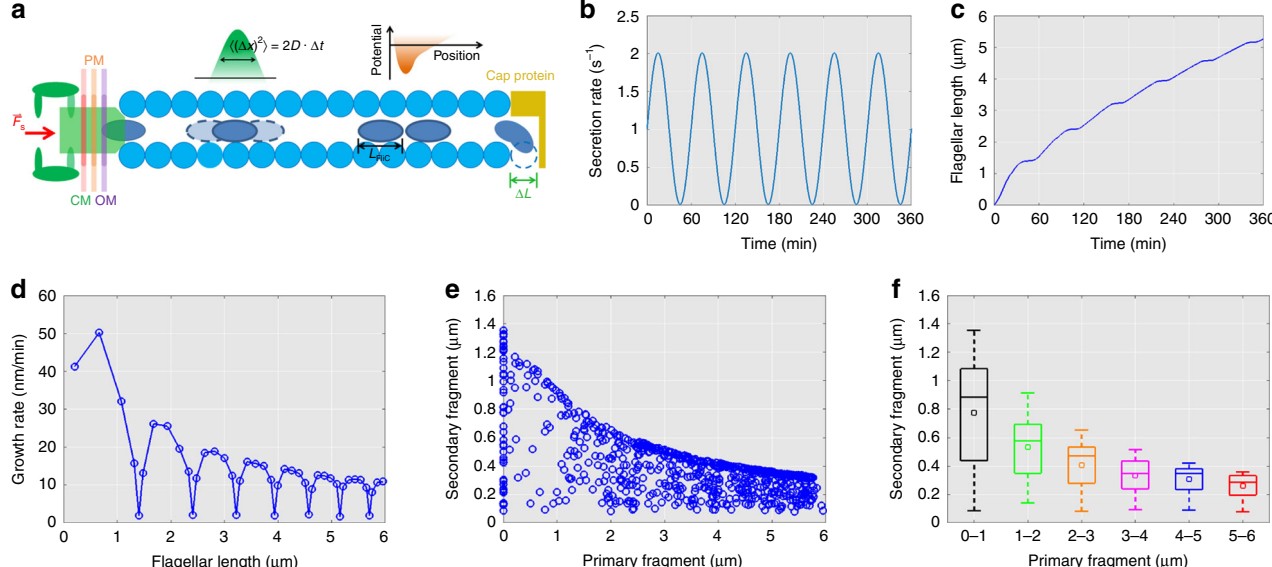

**Fig. 6** Single-file injection-diffusion model explains punctuated flagellar growth mechanism. **a** Schematic of single-file injection diffusion model. We first assume that T3FSS exerts a force on unfolded flagellin until it completely enters the central tube. Once the unfolded flagellin is inside the tube, it moves via diffusion. The single-file diffusion is modeled as one dimensional diffusion in the channel and the each flagellin subunit repels the other flagellin subunits, where encountered, via a Lennard-Jones potential. Finally, when an unfolded flagellin arrives at the distal end of the flagellum, it polymerizes onto the growing end of the filament immediately. **b** A simple sinusoidal variation form of the flagellar injection rate with a secretion variation period of $T_S = 60$ min. **c** The simulated flagellar length versus time and **d** flagellar growth rate versus flagellar length. **e** A population simulation with different variation period $T_S$ and variation phase $\varphi$ is plotted. **f** Box plot of **e** has been binned at a 1-μm intervals

the parameters that we used for the simulation can be found in the Methods.

In Fig. 6b, we show a typical time-dependent loading rate with $T_S = 60$ min. Using the model framework, the corresponding flagellar growth trajectory can be simulated, as demonstrated in Fig. 6c. Using the same calculation as in our experiments, the flagellar growth rate vs. length curve is plotted (Fig. 6d). The curve shows clear pause events during flagellar growth, along with the overall decay in flagellar growth rate, which is highly similar to our experimental observations (Figs. 2b, 3c and 4c, d). Here all pause events align with low flagellin secretion rate governed by the periodic fluctuation in FliC concentration.

We further assumed that the growth of different flagella follows the same mechanism but that the secretion variation phase may differ. We performed a coarse-grained search of model parameter space and used the magnitude of variation in flagellar growth rate (Figs. 3c and 4c, d), the scatter plots and mean value of secondary fragments vs. primary fragments in the double-color labeling experiment (Fig. 2b, c) to restrict the parameter space search. A parameter set was identified (Methods), and by varying the secretion variation phase $\varphi$ ($0$–$2\pi$), we generated many in silico flagellar growth trajectories. Following the experimental methods, we sampled the lengths of secondary fragments at various primary fragments ($0$–$6$ μm) and summarize the data in Fig. 6e. To enable a more precise comparison to the experimental data, the simulation data from Fig. 6e has been binned at a 1-μm intervals and visualized as box plots (Fig. 6f). The results reproduced two features of our experimental data: (1) the overall decay of the flagellar growth rate and (2) the high level of fluctuation in the growth rate. While assuming sinusoidal variation in FliC concentration is no doubt an oversimplification, our simulation results fit the experimental data for the secretion rate of flagellin very well. The exact model explaining the underlying variation in FliC concentration remains unknown, which warrants future experimental investigations.

## Discussion

In this work, by fluorescently labeling flagella with high temporal and spatial resolution, we were able to monitor *E. coli* flagellar growth in vivo and in real-time. The growth rate measurements over a population of cells showed that the rate of flagellar growth displays large fluctuation and, on average, decreases with length. Through the real-time tracking of single flagellar growth, we confirmed the large fluctuation in individual flagellar growth rates and discovered that the flagella of *E. coli* grew with frequent pauses. Our data are consistent with the earlier work by Turner et al.[18], who reported that *E. coli* flagellar growth rate show large fluctuation with a mean rate of ~13 nm/min. However, they did not report a decay in flagellar growth rate when flagella are long. We speculate that the shearing forces from repeated centrifugation and the long-term incubation at 7 °C used in their experiment[18] may have perturbed flagellar growth and that may explain why decay in growth rate of *E. coli* was not observed.

*S. enterica* and *E. coli* are close relatives in bacterial taxonomy and both have peritrichous flagella, yet their flagellar growth show distinct features. First, the initial growth rate of ~100 nm/min in *S. enterica*[22] is two times faster than that of *E. coli*, which has an initial growth rate of ~30 nm/min. Second, frequent pausing is common phenomena in *E. coli* flagellar growth, but not in *S. enterica*, where only a small fraction of flagella stopped growing and in those cases never resumed growing[22]. These differences between *E. coli* and *S. enterica* are possibly due to more subtle differences between these bacterial species and more experimental studies are needed to determine exactly which elements of flagellar export have changed.

Recently, to explain the constant flagellar growth rate of *E. coli*, a chain mechanism was proposed[19]. By introducing cysteines into FliC proteins, Evans et al.[19] provided evidence supporting a possible head-to-tail linkage of unfolded flagellin subunits in the transit channel. In this model, a continuous chain is formed in the channel through successive head-to-tail linkages connecting flagellin monomers and the entropic force generated from

crystallization of a flagellin monomer at the growing tip pulls the entire flagellin chain through the channel. However, the model failed to explain the decay in flagellar growth rate when flagella are long, as demonstrated by our high spatial and temporal resolution observation.

In addition, the chain model is not compatible with the frequent pauses in flagellar elongation. The working mechanism of the chain model predicts a smooth and continuous growth of the flagella. Only when the flagellin chain breaks in the channel does the apical piece continue to crystallize while leaving the basal piece stuck in the channel. In this case, the flagellar growth stops until the broken chain reaches the growing tip again. However, it is unknown what force would drive the remaining chain fragment to the tip. While the chain model allows flagellar growth to be discontinuous, it struggles to explain the relatively quick recovery in growth rate of E. coli in a 10-min time interval observed in our experiments. Moreover, the latest experimental evidence from Renault et al.[22] suggests that the requirements of head-to-tail linkage formation of flagellins in the channel is unreachable based on the known biophysical properties of flagellum assembly.

Discontinuous growth of bacterial flagellar filaments has been observed once before in vitro[33] using a dark-field light microscope to record the polymerization of monomeric flagellins. Each filament was observed to grow at a unique rate with occasional pausing in elongation. The attachment of inactive flagellins to the tip was proposed as the reason for these stoppages, since, in these in vitro experiments, the preparation of flagellin protein may produce denatured or impure flagellin monomers. Theoretically, such production of defective flagellin molecules during flagellar growth is possible, but it should occur infrequently and not be unique to a specific bacterial genus. In contrast, paused growth has not been observed in in vivo measurements of V. alginolyticus and S. enterica flagellar growth[21, 22]. In our experiments here, when we increased cytoplasmic flagellin expression we observed fewer pause events in flagellar growth. Given this, we believe that the frequent pauses we observed in E. coli flagellar growth were not caused by the terminal addition of inactive flagellin proteins.

In our effort to explore the potential factors causing pauses in flagellar growth, we found that the growth of flagella on single cells show similar growth patterns, which suggests that the growth rates of different flagella on the same cell are likely to be regulated by a global factor. Although the time points of real-time fluorescence tracking are limited, we calculated the Pearson's correlation coefficient (PCC) using flagellar growth trajectories for a quantitative description of the similar growth pattern between multiple flagella on the same cell. The correlation coefficient between two flagella in Fig. 4c is 0.953 and the correlation coefficients between each two flagella in Fig. 4d are 0.942 (red vs. blue), 0.906 (green vs. blue), and 0.805 (red vs. green), indicating that the flagellar growth processes were correlated. In the flagellin overexpression assay, we observed that increasing the expression levels of flagellin proteins reduced the ratio of pause events, implying that substrate insufficiency is an important factor for the continuous growth of E. coli flagella. Although we provided evidence that flagellin concentration contributes to pausing, we still observed ~20% of these pause events in our flagellin overexpression measurements. To investigate the subcellular localization and concentration of flagellins in bacteria, single-molecule super-resolution microscopy methods[34–36] are needed, as they provide the highest resolution. It has been reported elsewhere that the export efficiency of the flagellar type III secretion system depends on its energy source[8, 37]. The results of Renault et al.[22] showed low-speed flagellar growth after Carbonyl cyanide m-chlorophenyl hydrazone (CCCP) treatment which reduced the PMF. Therefore, it would be interesting to simultaneously monitor flagellar growth as well as change in cellular ATP concentration and PMF using fluorescent probes[38–40] to provide a comprehensive understanding of the molecular energetics accompanying flagellar growth.

In this work, we implemented a similar injection-diffusion model that was used before[21] to simulate flagellar growth in E. coli. The physical principle of the injection-diffusion model is one-dimensional Brownian diffusion model of flagellins with a pushing base and an absorbing tip. By varying parameters of the model, it can predict various growth rates vs. length dependence, which implies the flagellar growth of different bacterial species share similar physical principles but that the biological details may differ. We expect the model to apply more widely to extracellular protein translocation in other systems.

## Methods

**Construction of E. coli strains.** E. coli K 12 strain RP437 (CGSC 12122) with a tetracysteine tag (TC tag) insertion (RP437-TC-FliC strain) was used in this study for flagellar growth measurement. The RP437-TC-FliC strain was constructed by inserting the optimized TC tag (FLNCCPGCCMEP) with a chloramphenicol resistance gene and ccdB under control of the rhamnose promoter into the chromosomal wild-type fliC gene using the lamda-Red system. Specifically, the TC tag was incorporated after the proline residue at amino acid position 285 in the D-loop domain of the FliC protein[25]. To eliminate the antibiotic resistance and ccdB, the construct was selected by survival on rhamnose minimal medium (100 mM potassium phosphate (pH 7.0) (11.2 g/l $K_2HPO_4$, 4.8 g/l $KH_2PO_4$), 15 mM $(NH_2)_2SO_4$, 1 mM $MgSO_4$, 2 mM $Fe_2$ $(SO_4)_3$) with 25 mM rhamnose (Sigma Aldrich) and MEM vitamins (×100, Invitrogen) and amino acids (×50, Invitrogen))[41, 42]. For the flagellin overexpression assay, TC-FliC was expressed on the plasmid pBAD18 in the strain RP437-TC-FliC under the control of the arabinose-inducible promoter, maintained by the addition of ampicillin (50 μg/ml) to the growth medium. The presence of the tetracysteine tag in the fliC gene was verified by colony PCR and sequencing. Strains and primers used are listed in Supplementary Tables 1 and 2.

**E. coli growth and flagella labeling.** The RP437-TC-FliC strain was grown from frozen stocks (made from single colonies) in LB medium (1% w/v Bacto-Tryptone, 0.5% w/v yeast ex-tract and 1% w/v NaCl) overnight at 30 °C. The saturated culture was diluted by 1:100 in M9 medium (M9 minimal medium with 0.4% glucose, MEM vitamins (×100) and amino acids (×50)) and grown at 30 °C to an appropriate concentration ($OD_{600}$ = 0.3–0.4). Labeling of the flagella was performed by gently mixing FlAsH or ReAsH (Invitrogen, 2.5 μM) with the diluted E. coli cells and incubated for 10 min at 30 °C. For flagellar growth observations in the flagellin over-expression assay, M9 medium was supplemented with 0.5% glycerol, MEM vitamins and amino acids and ampicillin (50 μg/ml).

**Sample preparation.** Microscope tunnel slides were constructed by attaching double-sided tape between microscope slides and coverslips (Fisher Scientific), which were coated with 0.01% w/v poly-L-lysine (Sigma) for 1 min then rinsed with M9 medium. The stained bacterial cells were added into the tunnel and allowed to settle for 10 min in the dark. Unattached cells were removed by a M9 medium wash. For the real time observation of flagellar growth, 2.5 μM FlAsH was kept in M9 medium to ensure timely labeling of newly grown flagellar segments.

**Double-color and triple-color imaging by super-resolution microscopy.** It is reported that the average flagellar length of a bacterial population increases with the culturing time[16, 21]. Therefore, we cultured E. coli cells in M9 medium for 2 h, 4 h, and 6 h to achieve a wide range of primary flagellar lengths. Then, the culture was diluted to an appropriate concentration and labeled with 2.5 μM ReAsH. After centrifugation for 5 min at 1400 × g, cells were grown in fresh M9 medium at 30 °C for 30 min in the presence of 2.5 μM FlAsH. For triple color labeling, FlAsH was used twice to stain primary and tertiary segments of flagella, and ReAsH was used for the secondary segments. Two 30-min incubations were applied for the final two labeling periods. After labeling, super-resolution imaging was carried out by the Structured Illumination Microscopy (Nikon N-SIM) to achieve a better resolution picture of E. coli flagella. N-SIM was equipped with an EMCCD camera (iXon DU-897E), a 100 × 1.49 NA objective lens (CFI Apo TIRF ×100), a 488 nm excitation laser (Coherent Sapphire 488 LP), and a bandpass emission filter (500–550 nm, Chroma). Super-resolution images of E. coli flagella were reconstructed by NIS-Elements viewer 4.20.

**Real-time imaging by spinning disk confocal microscopy.** The real-time fluorescence imaging of flagellar growth was performed on an UltraVIEW VoX (Perkin Elmer) spinning disk confocal microscope. Specifically, a Nikon Ti-E Eclipse inverted microscope was equipped with a 1.4 NA 100 × CFI Plan Apo VC oil objective, an EMCCD camera (Hamamatsu), a Coherent solid state 488 nm/50 mW diode laser, a 527 nm bandpass center wavelength emission filter and the Perfect

Focus system. The prepared tunnel containing stained cells were set in a humidity chamber with a temperature-control system to allow flagellar growth at 30 °C. Time-lapse fluorescence images were captured at 10-min intervals using Volocity 3D Image Analysis Software running on a PC laptop. A 1.2 μm Z-axis scan, with images taken every 0.2 μm for each layer, was adopted to obtain high quality images of the bacterial flagella.

**Image analysis and flagellar growth rate measurements**. Flagellum analysis and the measurement of flagellar length were performed using the ImageJ software[43]. Since flagella were all curved, freehand lines were introduced into the fluorescence images to mimic the trace of flagella and fit spline tool was used to perfect the fittings. For images acquired by the spinning disk confocal microscope, the physical distance corresponding to one pixel is 144 nm and the flagellar length of 2D projection can be calculated accordingly. According to the previous study[21], the 2D-3D correction of flagellar length is minor, as the difference between them is <10%, so the flagellar length of 2D projection was used in our results. The difference between flagellar lengths measured at adjacent time points was considered as the newly grown length, and the flagellar growth rate was calculated as the newly growth length divided by the time interval (10 min). Compared with the whole period of flagellar growth, the measurement interval (10 min) was a short unit, and the average growth rate could be considered as the instantaneous rate at the midpoint of the newly grown length between two adjacent flagellar lengths.

By surveying a single flagellum several times with the ImageJ free-hand line tool, the obtained standard deviation of flagellar length measurements was approximately 50 nm. Thus length changes <50 nm during a 10-min interval can be considered as measurement error. Therefore, flagellar growth rates <5 nm/min is designated a pause event in flagellar growth tracking.

**FliC overexpression assay**. E. coli cells of two groups, RP437-TC-FliC and RP437-TC-FliC carrying plasmid pBAD18-TC-FliC supplemented with 0.2% arabinose, were grown in LB (the latter containing ampicillin) at 30 °C for 5 h. Afterwards, the cultures were heated at 65 °C for 5 min and harvested by centrifugation. The obtained cell pellets were then resuspended in Western Sample Buffer (950 μl 2 × Laemmli Buffer, Bio-Rad; 50 μl 2-Mercaptoethanol, Sigma) and heated at 95 °C for 10 min. Samples were normalized by cell density. After separated by SDS–PAGE gel and transferred to a PVDF membrane (Bio-Rad), samples were blocked in QuickBlock™ Buffer (Beyotime) for 1 h at room temperature. Primary incubation was with an anti-FliC rabbit antibody (Abcam, ab93713) diluted 1:10,000 or an anti-MreB rabbit antibody (a gift from Zhen Liu) diluted 1:5000 in QuickBlock™ Primary Antibody Dilution Buffer (Beyotime) overnight at 4 °C. Secondary incubation was with goat anti-rabbit HRP conjugate (Beyotime) diluted 1:2000 in QuickBlock™ Secondary Antibody Dilution Buffer (Beyotime) for 1 h at room temperature. Blots were treated with an ECL plus detection kit (Beyotime). Three independent experiments were performed.

**Statistical analysis**. Two-tailed Student's t-tests and Mann–Whitney tests were performed to determine significance. A statistically significant difference was defined as *$P < 0.1$, **$P < 0.05$, and ***$P < 0.01$.

**Brownian dynamics simulation of flagellar growth**. Following the model framework introduced in the main text, a custom MATLAB program that implemented Brownian dynamics simulation was developed to generate flagellar growth sample traces. In each simulation step, every flagellin monomer in the channel is polled according to Equation 10 to update its location in the channel. Each sample trace is simulated until the flagellar length reaches 6 μm. To balance numerical accuracy and computational load, the simulation time step $\Delta t = 10^{-4}$ to $10^{-6}$ s was chosen. We tested simulations with smaller $\Delta t$ and confirmed the numerical convergence of the simulated results.

Using this program, the movement of each flagellin molecule can be tracked, and flagellar growth can be simulated with high temporal resolution. Following the experimental procedure for calculating the instantaneous flagellar growth rate, the model can predict the relationship between flagellar growth rate and flagellar length. The parameter set that we used to generate results in the main text is as follows: $F_s = 0.5$ pN; Flagellin diffusion coefficient ($D$) = 5000 nm$^2$s$^{-1}$; $\varepsilon = 1$ $K_BT$; $\sigma = 74$ nm; $\Delta t = 0.0001$ s; $L_{FliC} = 74$ nm; $\Delta L = 0.47$ nm; $\bar{R} = 1.01$ s$^{-1}$; $R_0 = 1$ s$^{-1}$.

**Code availability**. The modeling code is available from the corresponding author upon request.

**Data availability**. The data that support the findings of this study are available in this article and its Supplementary Information Files, or from the corresponding authors upon request.

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

## Acknowledgements

This work is financially supported by the National Natural Science Foundation of China (Nos. 31722003,31770925, and 31327901) and the Recruitment Program of Global Youth Experts (to F.B.); the Ministry of Science and Technology, Republic of China under Contract No. MOST-103-2112-M-008-010-MY3/MOST-106-2112-M-008-023 (to C.L.); C.L. and F.B. are also supported by the Human Frontier Science Program Grant (RGP0041/2015). We also thank Hongxia Lv and Chunyan Shan in the Core Facilities of Life Sciences, Peking University, for assistance with spinning disk and SIM imaging.

## Author contributions

Z.Z., C.L., and F.B. designed research; Z.Z. and Y.Z. performed experiments; Z.Z., Y.Z., X. Z., W.L., and M.B. analyzed data; Z.Z., X.Z., W.L., M.B., C.L., and F.B. wrote the paper.

## Additional information

**Competing interests:** The authors declare no competing interests.

