## [Peer Review File · Nature Communications]

Reviewers' comments:

Reviewer #1 (Remarks to the Author):

This is an extensive multi-color biarsenical-dye-based study of flagellar-filament growth in *E. coli*. The work appears sound, but I have a few reservations, given below. Filament growth rates decline at large lengths, on average, but are discontinuous, interrupted by frequent pauses. Growth is modeled by an injection-diffusion mechanism, and the pauses are thought to arise from variations in cytoplasmic concentrations of flagellin.

Is Nature Communications concerned about the accessibility of work by the general reader? This is not a paper for the novice. Would a specialty journal like Nature Microbiology be better?

Pauses in the growth of flagellar filaments have been studied long ago *in vitro*, using purified flagellin from *Salmonella*: Micro-video study of discontinuous growth of bacterial flagellar filaments *in vitro*, Ishihara A, and Hotani, H, *J Mol Biol* 139, 265-276 (1980). Growth might be suspended for many minutes and then resume. It was suggested that a defective flagellin molecule might be incorporated at the end of a growing filament and block further growth. Eventually, when that subunit managed to fall off, growth could continue. Couldn't this happen *in vivo*? Growth might cease until the defective subunit came off and was passed into the external medium by the distal cap, FliD.

It is known that hook-associated proteins, including FliD, are found in the external medium during normal cell growth. The leakage of FliD makes it possible for a filament to grow again after it has broken. Flagellin is dumped into the external medium until enough FliD comes along to reconstitute the cap, and then polymerization of flagellin continues. In my mind, this is the main reason that the chain mechanism for flagellar growth (ref 19) fails: if a filament breaks, what pulls the remaining chain of flagellin out of the proximal segment of the filament?

The pore down the axis of a flagellar filament is about 2 nm in diameter. How can a sphere of diameter LFliC = 74 nm (pp. 15 and 26) be accommodated?

Reviewer #2 (Remarks to the Author):

Flagella are used for swimming by bacteria. The flagellum is assembled from the inside out, beginning with the basal body embedded in the cell membrane. The structural components of the sequential rod, hook and filament are unfolded and exported through a type III flagellar secretion system (T3FSS). Conflicting evidence exist whether the rate of flagellar synthesis is directly related to their length. This paper describes the measurement of the rate of flagellar synthesis of individual flagella using a dye that binds to 4 cysteines that are engineered into the flagellar subunits. The introduction of the cysteines and the staining with the dye did not affect the cell growth or the number or length of flagella on the cells. It is observed that the growth of the flagella is inversely correlated with their length. That individual flagella have different growth rates and that they have also different growth rates regimes during their growth. The growth pauses frequently. Flagella on one cell behave more or less the same. An injection-diffusion-crystallization model is adapted to explain the growth behavior when the parameters are varied. The paper is well written and the figures are very clear.

Major comments:

Poly lysin is used to immobilize the cells between the object and the coverglass. It is positively charged. The FliC protein has quite a number of negative charges (54) and less positive charge (37).

Could it be that the frequent pauses in flagellar growth are caused by binding temporarily to the poly-Lys? Would it not be possible that the longer the flagella the more often it binds to the poly-Lys hence the reduction in the rate of lengthening? Binding to poly-Lys would also explain the variability between the various flagella as it is a stochastic process. Maybe it is not at all true, but it would be very supportive if the authors could do a similar experiment with cells that are immobilized on an agarose pad for instance. This will not yield such nice images, but as a supplementary figure it would help to avoid the notion that the behavior of the filaments could be affected by the immobilization method.

Turner et al. 2012 J. Bacteriol observed constant growth rate of the flagella of *E. coli*. He also used poly-Lys but he analyzed the flagella of fixed bacteria that has been dyed during growth.

Chen et al., 2017 Elife uses also poly-lysine coated object glasses and writes "Because the sheathed flagella readily attach to the poly-L-lysine coated surface, both the cell body and its flagellum are well immobilized on the microscope tunnel slides, enabling several hours of observation to track the flagellar growth. Time-lapse fluorescent images were captured at 5/20 min intervals depending on the flagellar growth rate." This is on *Vibrio alginolyticus* flagella. They also write "We observed that the flagellar growth rate of *V. alginolyticus* is highly length-dependent, with distinct growth regimes. Initially, the flagellum grows at a constant rate when its length is below 1500 nm. While it continues to grow, the growth rate decreases sharply." The distinct growth regimes can also be influenced by the poly-lysine.

Renault, T. T. et al. Bacterial flagella grow through an injection-diffusion mechanism. *eLife* 6 (2017). This is on *Salmonella enterica* serovar Typhimurium. They use also poly-Lys, but fixed the cells.

Overall, I think that the present paper is the first to describe the growth of individual flagella while being attached to the poly-lys surface. Therefore, it seems to me important to show that the growth of the flagella is not affected by the interaction with the poly-Lys.

The fact that flagella from a single cell had the same growth pattern, suggest that this is an intrinsic characteristic and not caused by the poly-Lys. But growth on a different surface will confirm this.

Overexpression of flagellin reduced the number of pauses in flagellar growth, suggesting that the number of available flagellin subunits is the limiting factor in growth. But this does not explain the relation between growth rate and flagellar length.

The authors create a model that fits the observed experimental behavior of the flagella growth, but they do not have an explanation for the fluctuations in the concentration of FliC which is required for the model.

The model requires Brownian motion of partly unfolded flagellin through the sheet, crystallization of flagellin at the end and pushing from the base by inserting new subunits, which requires ATP and the PMF. A similar model was suggested by Chen et al 2017 Elife.

The author discuss why other models are likely not correct, but they do not discuss what the limiting factors are for their growth rate, pausing, variability within one flagellum and the relation between length and growth rate. They show that they can simulate the growth of the flagellar population but have no clear explanation for the behavior of the individual flagella.

Minor comments:

Methods:

The authors should mention between which amino acids the TC was inserted in the RP437-TC-FliC strain. TC-FliC is written, but the word insertion is used. Suggesting that it is not a N-terminal fusion....?

The chemicals are not written as is agreed internationally. What amino acids are added to the medium and in which concentration?

Reviewer #3 (Remarks to the Author):

This manuscript reports a detailed study of flagellar growth in *E. coli* at the single-cell level. The authors construct a flagellin derivative carrying an internal tetracysteine tag that can be specifically labeled with the green or red fluorescent dyes FIAsh and ReAsH, respectively. Using this tool, they perform two-color labeling experiments in which cells are exposed to alternating pulses of these dyes. A quantification of the lengths of the labeled flagellar segments indicates that the rate of flagellar growth decreases with increasing flagellar length. Live-cell imaging then reveals that flagellar growth is interrupted by frequent pauses that occur independently of the length of the flagellar filament. Interestingly, the authors report a correlation in the growth rates and in the timing and length of pause events for flagella that emerge from the same cell. Based on the observation that overproduction of flagellin appears to shorten the duration of pauses, the authors postulate that pause events may reflect variations in the cellular pool of flagellin subunits. Finally, the authors implement an injection-diffusion model for flagellar growth that includes variations in the flagellin, which is able to reproduce the experimental results.

This paper nicely shows that the dynamics of flagellar growth in *E. coli* are similar to those previously measured for related species. In addition, it reports the novel observation that flagellar assembly is interrupted by pauses that may result from fluctuations in the flagellin level. These latter findings are exciting and may shed new light on the process of flagellar biosynthesis. However, there are several issues that need to be addressed before publication to support the conclusions drawn.

Major issues:

1. The authors provide evidence that the presence of FIAsh does not affect cell growth (Supplementary Figure 2). However, the two-color labeling of successive flagellar segments involves washing steps. The authors should provide evidence showing that these washing steps do not have any adverse effects on the physiology of the cells that may reduce the rate of flagellar growth.
2. The authors state that it is not possible to determine a precise correlation coefficient for the growth rates and for the timing and duration of pauses of multiple flagella emerging from the same cell (Figure 4c and d). However, the uniform behavior of these flagella is one of the key points of this paper. Therefore, the authors should make an effort to support their conclusions with more quantitative data. In the present examples, the growth rates of the flagella compared do indeed appear similar. However, there are time points during which some flagella grow whereas others do not. Thus, it remains unclear how uniform the behavior of flagella really is.
3. The data for the uninduced and induced cells in Figure 5c do apparently not result from the same experiment (see line 260). How reproducible are these measurements? Can the authors exclude the possibility that the differences observed are due to slight differences in the growth conditions?

4. For the real-time measurements of flagellar growth, the authors use cells that are immobilized on poly-lysine-coated slides. It is well-established that poly-lysine is toxic to bacteria, both in solution and as surface coating (e.g. Colville et al, Langmuir, 2010, 26: 2639). How did the authors ensure that the physiology of the cells was not affected under the conditions used? What were the growth rates of the cells in the tunnel slides? Is it possible that the pauses observed in this study are due to poly-lysine-induced cell damage? Are similar data obtained when cells are immobilized mechanically (e.g. in CellASIC ONIX flow cells, as used by Renault et al, 2017)?

5. The major new aspect of the injection-diffusion model developed by the authors is the implementation of fluctuations in the flagellin concentration. However, although the model can reproduce the experimental results, it remains unclear whether these fluctuations actually exist. Therefore, the value of the model remains unclear.

Minor issues:

6. In the examples given in Figure 4a and b, the flagellar growth rates are below the population average. Is this true for all cases in which cells display multiple flagella? Are there also examples of cells carrying multiple flagella that all grow at rates higher than the population average? Are the growth rates measured in this case as similar as for the examples shown in Figure 4a and b?

7. In cells carrying the *fliC* overexpression plasmid (Figure 5b), the increase in the level of flagellin is surprisingly low, considering that the promoter system employed is routinely used for protein purification purposes. Do the authors have any explanation for that?

8. What is the total number of FliC molecules in a wild-type cell? Is this number really limiting, considering the number of molecules required to synthesize flagella at the rates measured in this study?

9. The statement that bacteria live in environments of low Reynolds numbers refers to the motion of bacterial cells in liquid solutions. It is not clear to me why this should also apply to the motion of proteins encapsulated in the flagellar filament (line 327).

10. The authors state the rate at which new flagellin subunits are loaded by the export apparatus is proportional to the flagellin concentration. Do they assume a linear relationship? Shouldn't the rate depend on the affinity (K_d value) of the export apparatus for its substrate and thus show a non-linear dependency on the FliC concentration?

Comments:

11. line 67: "with an optimized"

12. line 68: "with biarsenic dyes"

13. lines 96ff: "tetracysteine tag (...) -containing"

14. throughout the manuscript: "spinning-disc confocal microscopy"

15. line 126: "showed large variations"

16. throughout the paper: "fluorescence microscopy/imaging".

17. line 163: Why "in contrast"? This study leads to the same conclusion. It additionally suggests that there may be frequent pauses.

18. line 182: "growth of the"

19. lines 185-188: This part should be moved to the figure legend.

20. line 242: "expression of a plasmid-borne RP437-TC-*fliC* gene".

21. Supplementary Figure 4: Did both strains contain the overexpression plasmid or only the strain that was grown in the presence of arabinose?

22. line 257: "on flagellar growth"
23. line 263: "recovered by"
24. line 355: "follows"
25. line 395: "between"
26. References: Check for italicization of gene and species names, small print of titles, and proper abbreviation of journal names (e.g. references 7, 12, 15, 18, 20, 24, 25, 29, 31, 36).

Reviewer 1

This is an extensive multi-color biarsenical-dye-based study of flagellar-filament growth in *E. coli*. The work appears sound, but I have a few reservations, given below. Filament growth rates decline at large lengths, on average, but are discontinuous, interrupted by frequent pauses. Growth is modeled by an injection-diffusion mechanism, and the pauses are thought to arise from variations in cytoplasmic concentrations of flagellin.

RESPONSE: We thank Reviewer 1 for the positive and constructive comments.

Is Nature Communications concerned about the accessibility of work by the general reader? This is not a paper for the novice. Would a specialty journal like Nature Microbiology be better?

RESPONSE: The bacterial flagellum is a unique biological proteinaceous structure and also a self-assembling regenerating nanomachine. We believe the general readers of Nature Communications would appreciate this work, particularly as our model for extracellular transport may apply to other secretion systems and transporters in general beyond microbiology.

Pauses in the growth of flagellar filaments have been studied long ago *in vitro*, using purified flagellin from Salmonella: Micro-video study of discontinuous growth of bacterial flagellar filaments *in vitro*, Ishihara A, and Hotani, H, J Mol Biol 139, 265-276 (1980). Growth might be suspended for many minutes and then resume. It was suggested that a defective flagellin molecule might be incorporated at the end of a growing filament and block further growth. Eventually, when that subunit managed to fall off, growth could continue. Couldn't this happen *in vivo*? Growth might cease until the defective subunit came off and was passed into the external medium by the distal cap, FliD.

RESPONSE: We followed reviewer's suggestion to discuss more about the possibility that flagellar growth might cease when an inactive flagellin attaches to the tip (Line 416-429 in the revised manuscript). For the *in vitro* experiment (Ref 32), the rest phase might occur when a partially denatured flagellin attaches. Theoretically, the production of defective flagellin molecules during flagellar growth is possible but should occur at a low frequency and not be unique to any specific bacterial genus. In contrast, paused growth was not observed in *in vivo* measurement of *V. alginolyticus* and *S. enterica* flagellar growth (Ref 21 and 22). In our experiments, when we increased cytoplasmic flagellin expression we observed far fewer pause events in flagellar growth. Given this, although we cannot completely rule it out, we believe that the frequent pauses we observed in *E. coli* flagellar growth were not primarily caused by the terminal addition of inactive flagellin proteins.

It is known that hook-associated proteins, including FliD, are found in the external

medium during normal cell growth. The leakage of FliD makes it possible for a filament to grow again after it has broken. Flagellin is dumped into the external medium until enough FliD comes along to reconstitute the cap, and then polymerization of flagellin continues. In my mind, this is the main reason that the chain mechanism for flagellar growth (ref 19) fails: if a filament breaks, what pulls the remaining chain of flagellin out of the proximal segment of the filament?

RESPONSE: We agree with the reviewer that the chain mechanism is problematic. The chain mechanism was proposed to explain the constant rate of flagellum growth, in which the entropic force derived from flagellin crystallization automatically adjusts with the length of the multi-subunit chain to support a constant rate of subunit transit that is independent of channel length. However, as Reviewer 1 had noticed, in the case where the multi-subunit chain breaks, the chain of distal subunits is pulled to the growing tip, leaving the chain of basal subunits to diffuse in the channel. Under this circumstance, it is unknown what force would pull the remaining chain out of the proximal segment of the filament to continue flagellar growth. For this point, we share the same opinion as Reviewer 1.

The pore down the axis of a flagellar filament is about 2 nm in diameter. How can a sphere of diameter $L_{FliC} = 74$ nm (pp. 15 and 26) be accommodated?

RESPONSE: We thank the reviewer for raising this question. We are sorry for being unclear in the manuscript. In the one dimensional simulation, we assumed that the flagellin subunits in the channel are partially unfolded mainly as α -helical chains, which are ~ 1 nm in diameter by 74 nm long (Ref 23) and we have clarified this in our revised manuscript (Line 320). The 'sphere' is not an actual structure of the unfolded flagellin. In our simulation, we modeled that the force between any pair of unfolded flagellin monomers can be calculated by the Lennard-Jones potential, which assumes the range of potential is symmetric around the mass center of each flagellin monomer. In fact, our model was based on a one dimensional simulation, therefore we admit that 'spherical' is a misleading term. We have corrected it in our revised manuscript (Line 321-322).

Reviewer 2

Flagella are used for swimming by bacteria. The flagellum is assembled from the inside out, beginning with the basal body embedded in the cell membrane. The structural components of the sequential rod, hook and filament are unfolded and exported through a type III flagellar secretion system (T3FSS). Conflicting evidence exist whether the rate of flagellar synthesis is directly related to their length. This paper describes the measurement of the rate of flagellar synthesis of individual flagella using a dye that binds to 4 cysteines that are engineered into the flagellar subunits. The introduction of the cysteines and the staining with the dye did not affect the cell growth or the number or length of flagella on the cells. It is observed that the growth of the flagella is inversely correlated with their length. That individual flagella

have different growth rates and that they have also different growth rates regimes during their growth. The growth pauses frequently. Flagella on one cell behave more or less the same. An injection-diffusion-crystallization model is adapted to explain the growth behavior when the parameters are varied. The paper is well written and the figures are very clear.

RESPONSE: We thank Reviewer 2 for the nice summary and very positive comments.

Major comments:

Poly-lysine is used to immobilize the cells between the object and the cover glass. It is positively charged. The FliC protein has quite a number of negative charges (54) and less positive charge (37). Could it be that the frequent pauses in flagellar growth are caused by binding temporarily to the poly-Lys? Would it not be possible that the longer the flagella the more often it binds to the poly-Lys hence the reduction in the rate of lengthening? Binding to poly-Lys would also explain the variability between the various flagella as it is a stochastic process. Maybe it is not at all true, but it would be very supportive if the authors could do a similar experiment with cells that are immobilized on an agarose pad for instance. This will not yield such nice images, but as a supplementary figure it would help to avoid the notion that the behavior of the filaments could be affected by the immobilization method.

Turner *et al.* 2012 J. Bacteriol observed constant growth rate of the flagella of *E. coli*. He also used poly-Lys but he analyzed the flagella of fixed bacteria that has been dyed during growth.

Chen *et al.*, (2017, eLife) uses also poly-lysine coated object glasses and writes “Because the sheathed flagella readily attach to the poly-L-lysine coated surface, both the cell body and its flagellum are well immobilized on the microscope tunnel slides, enabling several hours of observation to track the flagellar growth. Time-lapse fluorescent images were captured at 5/20 min intervals depending on the flagellar growth rate.” This is on *Vibrio alginolyticus* flagella. They also write “We observed that the flagellar growth rate of *V. alginolyticus* is highly length-dependent, with distinct growth regimes. Initially, the flagellum grows at a constant rate when its length is below 1500 nm. While it continues to grow, the growth rate decreases sharply.” The distinct growth regimes can also be influenced by the poly-lysine.

Renault, T. T. *et al.* Bacterial flagella grow through an injection-diffusion mechanism. eLife 6 (2017).

This is on *Salmonella enterica* serovar Typhimurium. They use also poly-Lys, but fixed the cells.

Overall, I think that the present paper is the first to describe the growth of individual flagella while being attached to the poly-lys surface. Therefore, it seems to me important to show that the growth of the flagella is not affected by the interaction with

the poly-Lys.

The fact that flagella from a single cell had the same growth pattern, suggest that this is an intrinsic characteristic and not caused by the poly-Lys. But growth on a different surface will confirm this.

RESPONSE: We thank the reviewer for this suggestion. To further prove that the observed pauses in flagellar growth were not an artifact caused by flagella attaching to poly-l-lysine on the surface, we followed the reviewer's advice and repeated the experiment with cells immobilized on an agarose pad while the flagella were free to move in space. The results are shown in the figure below (also added to Supplementary Information Line 72-85 and Supplementary Fig.4). Because of the stronger background on the agarose pad and the movement of flagella, there were only a limited number of cells in which flagellar length could be measured accurately. Supplementary Fig.4a displays two representative examples of single flagellar growth on the agarose pad with 10-min time interval. The growth measurement of 9 individual flagella is also shown in Supplementary Fig.4b-d. We see that although we have a limited amount of data, we are still able to observe pausing in flagellar growth, confirming our earlier conclusion that it is an intrinsic feature of *E. coli* flagellar growth.

Supplementary Fig. 4 Flagellar growth rate measured from *E. coli* cells immobilized using the agarose pad. (a) Representative images of two single growing bacterial flagellum of cells immobilized on the agarose pad, taken at time points with 10-min time interval. Scale bar, 1 μ m. **(b/c)** Growth measurements of 9 individual flagella tracked by real-time imaging. Plots of flagellar length versus time (b) and the flagellar growth rates versus flagellar lengths (c) are displayed. The flagella in (a) are emphasized by orange line with triangles (Top) and purple line with squares (Bottom). **(d)** Distribution of pause events (Blue) occurring in the total collected data points (Dark gray) at different flagellar lengths, binned at 200-nm intervals. All data points are from the measurements in (c).

Overexpression of flagellin reduced the number of pauses in flagellar growth, suggesting that the number of available flagellin subunits is the limiting factor in growth. But this does not explain the relation between growth rate and flagellar length.

RESPONSE: In our double- and triple-color labeling experiments, we show that on average, there is a decay in flagellar growth rate in respect to its length. This can be explained by the injection-diffusion model. When the flagellum is short, the flagellin inserted at the base would quickly diffuse to the tip. Only when the flagellum grows longer does diffusion of flagellin become the rate-limiting step, which reduces the growth rate. As Reviewer 2 mentioned, in our experiment, the number of available flagellin subunits is found to affect the frequency of pauses in flagellar growth. Considering that pauses in growth were observed to occur at different flagellar lengths, the decay in average speed with length is not caused primarily by an increase in the number of pause events.

The authors create a model that fits the observed experimental behavior of the flagella growth, but they do not have an explanation for the fluctuations in the concentration of FliC which is required for the model.

RESPONSE: We thank the reviewer for raising this issue.

1) During assembly of flagellar filaments outside the cell, there is continuous consumption of flagellin proteins inside the cell. When the production of new flagellin protein inside the cell does not provide enough flagellin to maintain growth at a certain rate then the rate diminishes until the concentration of FliC is restored.

2) We performed an additional experiment to check the concentration of intracellular flagellin. Time-lapse recording of cellular FliC-FIAsH fluorescence was carried out to validate the fluctuation in concentration of flagellin inside the cells. Briefly, the saturated culture of RP437-TC-FliC strain was diluted by 1:100 in M9 medium (M9 minimal supplemented with 0.4% glucose, 100x MEM vitamins, 50x amino acids and 10mM EDTA) and grown at 30°C to an appropriate concentration ($OD_{600}=0.3-0.4$). Labeling of intracellular flagellins was performed by gently mixing 5 μ M FIAsH with the diluted *E. coli* cells and incubating for 30 min at 30°C. Time-lapse imaging was performed to record cell fluorescence with 10-min time intervals and ImageJ software was applied to select bacteria for fluorescence intensity quantification. For the majority of bacterial cells, newly born flagella occurred in the focal plane during the observation, which could not be used for analysis. In our results, 15 cells were successfully measured and two typical examples of cell fluorescence recording were shown in the figure below. We observed that mean fluorescence intensity of cells exhibited variable changes with cell growth, indicating that flagellin density inside the cells underwent fluctuation.

Validation of the temporal fluctuation of intracellular flagellins. (a) Representative images of three fluorescent bacterial cells stained with FIASH and recorded by 10-min interval time-lapse imaging. Scale bar, 1 μm. (b) Plots of mean fluorescence intensity (arbitrary unit) of single cells versus time.

The model requires Brownian motion of partly unfolded flagellin through the sheet, crystallization of flagellin at the end and pushing from the base by inserting new subunits, which requires ATP and the PMF. A similar model was suggested by Chen *et al* (2017, eLife). The author discuss why other models are likely not correct, but they do not discuss what the limiting factors are for their growth rate, pausing, variability within one flagellum and the relation between length and growth rate. They show that they can simulate the growth of the flagellar population but have no clear explanation for the behavior of the individual flagella.

RESPONSE: We thank the reviewer for raising this concern.

In the current work, the flagellar growth pattern of *E. coli* is substantially different from that of *V. alginolyticus* in two aspects: 1) the flagellar growth rate displays large variations at similar lengths; 2) from single flagellar growth tracking we discovered that the large variation in growth rate occurs as a result of frequent pausing.

In our manuscript, we discussed the relationship between length and growth rate, variability and pausing events in single flagellum growth and in multiple flagella on the same cell. We searched the parameter space to reproduce the growth rate at different lengths for a population of cells. The reason why we did not put special emphasis on explaining the behavior of individual flagella is because: 1) on the experimental side, due to the technical difficulties in single flagellar growth tracking, we can only follow the growth of a limited number of flagella over a short time window. Therefore, we cannot rigorously quantify interval and frequency of pausing events, variability in growth speed at the single flagellum level; 2) on the modeling side, our injection-diffusion model includes many stochastic factors that contribute to the flagellar self-assembly process, such as the diffusion coefficient of flagellin monomers inside the channel, the loading speed of new flagellin monomers at the base, and different phases of the secretion process, many of which are not well characterized. In our current work, we hope to illustrate the unique features of *E. coli*

flagellar growth and present a simple mechanism which explains these features. Due to the challenges in performing long-time single flagellar growth tracking and the many factors affecting its growth, it is difficult for us to build a detailed model for the growth of an individual flagellum. By using a parameter set including periodic fluctuation in FliC concentration and variation in secretion phase we have been able to generate flagellar growth trajectories and reproduce our experimental data at the population level. We propose to leave the development of a detailed model for the growth of an individual flagellum to a future study, after the separate mechanisms of each component of flagellar export are more fully elucidated.

Minor comments:

Methods:

The authors should mention between which amino acids the TC was inserted in the RP437-TC-FliC strain. TC-FliC is written, but the word insertion is used. Suggesting that it is not a N-terminal fusion....?

RESPONSE: We thank the reviewer for raising this question. We are sorry for being unclear in the manuscript. In fact, the TC tag was incorporated after the proline residue at amino acid position 285 in the D-loop domain of the FliC protein (Ref 25) and we added this information in Line 468-470 of the revised manuscript.

The chemicals are not written as is agreed internationally. What amino acids are added to the medium and in which concentration?

RESPONSE: We thank the reviewer for considerate advice and we have corrected the writing of chemicals (Line 471-472) and added the information of amino acids and vitamins (Line 473 and line 482) in the revised manuscript.

Reviewer 3

This manuscript reports a detailed study of flagellar growth in *E. coli* at the single-cell level. The authors construct a flagellin derivative carrying an internal tetracysteine tag that can be specifically labeled with the green or red fluorescent dyes FAsH and ReAsH, respectively. Using this tool, they perform two-color labeling experiments in which cells are exposed to alternating pulses of these dyes. A quantification of the lengths of the labeled flagellar segments indicates that the rate of flagellar growth decreases with increasing flagellar length. Live-cell imaging then reveals that flagellar growth is interrupted by frequent pauses that occur independently of the length of the flagellar filament. Interestingly, the authors report a correlation in the growth rates and in the timing and length of pause events for flagella that emerge from the same cell. Based on the observation that overproduction of flagellin appears to shorten the duration of pauses, the authors postulate that pause events may reflect variations in the cellular pool of flagellin subunits. Finally, the authors implement an injection-diffusion model for flagellar growth that includes variations in the flagellin, which is able to reproduce the experimental results.

This paper nicely shows that the dynamics of flagellar growth in *E. coli* are similar to those previously measured for related species. In addition, it reports the novel observation that flagellar assembly is interrupted by pauses that may result from fluctuations in the flagellin level. These latter findings are exciting and may shed new light on the process of flagellar biosynthesis. However, there are several issues that need to be addressed before publication to support the conclusions drawn.

RESPONSE: We thank Reviewer 3 for the nice summary and very positive comments.

Major issues:

1. The authors provide evidence that the presence of FIA_SH does not affect cell growth (Supplementary Figure 2). However, the two-color labeling of successive flagellar segments involves washing steps. The authors should provide evidence showing that these washing steps do not have any adverse effects on the physiology of the cells that may reduce the rate of flagellar growth.

RESPONSE: We thank the reviewer for pointing this out. We followed the reviewer's suggestion to check the effects of washing steps on the physiology of bacteria. Briefly, we cultured *E. coli* cells in M9 medium for 4 hours and diluted the culture to an appropriate concentration and labeled with 2.5 μ M ReAsH. For the unwashed group, cells were imaged directly by bright-field microscopy to record bacterial growth for several hours. For the washed group, cells were washed by centrifugation for 5 min at 1400 x g then observed under the microscope. For each group, 100 cells were measured and the doubling time of the bacteria is shown in the figure below (also added to Supplementary Information Line 102-110 and Supplementary Fig.5c). No significant difference was found between the two groups, indicating that washing steps did not affect the physiology of the cells.

Supplementary Fig. 5 Examination of cell physiology. (c) Doubling time comparison of cells washed by centrifugation (Blue) or without washing (Red). Both washed and unwashed groups were performed on poly-L-lysine coated surface. 100 cells for each group were analyzed. Mean \pm SD: Red, 49.7 \pm 11.1 min; Blue, 49.5 \pm

13.7 min. Statistical analysis of two-tailed t-test was performed defining difference as insignificant, $P = 0.908$.

2. The authors state that it is not possible to determine a precise correlation coefficient for the growth rates and for the timing and duration of pauses of multiple flagella emerging from the same cell (Figure 4c and d). However, the uniform behavior of these flagella is one of the key points of this paper. Therefore, the authors should make an effort to support their conclusions with more quantitative data. In the present examples, the growth rates of the flagella compared do indeed appear similar. However, there are time points during which some flagella grow whereas others do not. Thus, it remains unclear how uniform the behavior of flagella really is.

RESPONSE: We appreciate this request. In our experiments, it is hard to follow continuous growth of flagella with high recording frequency due to the photobleaching effect of FIAsh. Additionally, the growth rate of *E. coli* is much slower (~13nm/min (Ref 18)) than other species like *V. alginolyticus* and *S. enterica* flagellar growth (~50nm/min (Ref 21 and 22)), making it difficult to obtain a minimal filament length required for visualization in less than 10 minutes. Although the time points are limited, we still followed the reviewer's suggestion to provide a quantitative description of the similar growth pattern between multiple flagella on the same cell (also added to Line 433-439). For any pair of flagella on the same cell, the Pearson's correlation coefficient (PCC) was calculated using their respective growth trajectories. In Figure 4c, the correlation coefficient between two flagella is 0.953, indicating that the flagellar growth processes were correlated. In Figure 4d, the correlation coefficients between each two flagella are 0.942 (red vs. blue), 0.906 (green vs. blue) and 0.805 (red vs. green). So we speculated that different flagella on the same cell are likely to be influenced by a global factor. This also supports the results from Figure 4a and 4b in which the growth rates of multiple filaments on the same cell were uniformly slower than the average growth speed of the population. As for the time points during which some flagella grow whereas others do not, we speculate that a delay might occur in growth rate in response to changes in FliC fluctuation as these flagella grow with different initial lengths.

3. The data for the uninduced and induced cells in Figure 5c do apparently not result from the same experiment (see line 260). How reproducible are these measurements? Can the authors exclude the possibility that the differences observed are due to slight differences in the growth conditions?

RESPONSE: We thank the reviewer for raising the question. We apologize for having been unclear. In the revision, we have made effort to repeat two conditions in one experiment. In brief, we cultured *E. coli* cells of two groups, RP437-TC-FliC and RP437-TC-FliC carrying plasmid pBAD18-TC-FliC supplemented with 0.2% arabinose, in M9 medium at 30°C for 4 hours and performed double-color labeling by ReAsH and FIAsh for 30min, successively. The measurements of three independent

experiments were shown in the figure below (also modified in Figure. 5c and d and Line 260-261 in the revised manuscript). Consistent with our former results, flagellar growth of the strain induced with arabinose was faster on average.

Figure 5 (c) and (d). (c) Scatter plot of the lengths of secondary versus the primary segments from which they grew for 30 minutes by SIM double-color imaging (For two cultures, RP437-TC-FliC strains with or without plasmid expression, 75 and 77 flagella were analyzed respectively.). (d) Average length of the secondary segments during 30 minutes for the two cultures in (c), binned at a 1-μm interval.

4. For the real-time measurements of flagellar growth, the authors use cells that are immobilized on poly-lysine-coated slides. It is well-established that poly-lysine is toxic to bacteria, both in solution and as surface coating (e.g. Colville *et al.*, Langmuir, 2010, 26: 2639). How did the authors ensure that the physiology of the cells was not affected under the conditions used? What were the growth rates of the cells in the tunnel slides? Is it possible that the pauses observed in this study are due to poly-lysine-induced cell damage? Are similar data obtained when cells are immobilized mechanically (e.g. in CellASIC ONIX flow cells, as used by Renault *et al.*, 2017)?

RESPONSE: We thank the reviewer for asking this issue.

(1) We note that although we both used 0.01% poly-L-lysine (PLL) solution from Sigma, the concentration of PLL coating in the Langmuir report is far higher. According to their study, they tried three types of PLL coating methods, including “rinse”, “drain”, and “dry” method. Even for “rinse” method which is believed to be thin PLL coating, the PLL solution was allowed to sit on the coverslip for 30-60 min and then washed. In our experiment, we only coated PLL for 1 min then rinsed with M9 medium. Therefore, their PLL coatings would prepare much thicker layer than ours, which would have a greater effect on bacterial physiology.

(2) We followed the reviewer's suggestion and checked if 1-min PLL coating adopted in our experiments had an effect on bacterial physiology. By growing cells on PLL coated coverslip and agarose pad, cell growth was recorded under bright-field microscope for several hours. The doubling time of cells grown on two surfaces were measured and results are shown in figure below (also added to Supplementary

Information Line 87-101 and Supplementary Fig.5a and b). No statistically significant difference in doubling time was observed between the two groups, indicating that 1-min PLL coating did not affect cell growth and viability.

Supplementary Fig. 5 Examination of cell physiology. (a) Time-lapse bright-field images of bacterial growth recorded on gel pad (Top) and poly-L-lysine (Bottom). Scale bar, 1 μm . (b) Doubling time comparison of cells grown on gel pad (Green) and poly-L-lysine (Purple). 100 cells for each group were analyzed. Mean \pm SD: Green, 50.8 ± 10.9 min; Blue, 53.3 ± 12.8 min. Statistical analysis of two-tailed t-test was performed defining difference as insignificant, $P = 0.439$.

(3) We thank the reviewer for this suggestion. To further prove that the observed pauses in flagellar growth were not an artifact caused by flagella attaching to poly-l-lysine on the surface, we followed the reviewer's advice and repeated the experiment with cells immobilized on an agarose pad while the flagella were free to move in space. The results are shown in the figure below (also added to Supplementary Information Line 72-85 and Supplementary Fig.4). Because of the stronger background on the agarose pad and the movement of flagella, there were only a limited number of cells in which flagellar length could be measured accurately. Supplementary Fig.4a displays two representative examples of single flagellar growth on the agarose pad with 10-min time interval. The growth measurement of 9 individual flagella is also now shown in Supplementary Fig.4b-d. We see that although the data points are not many, the pauses in flagellar growth rate were reproduced, confirming that pausing is an intrinsic feature of *E. coli* flagellar growth.

Supplementary Fig. 4 Flagellar growth rate measured from *E. coli* cells immobilized using the agarose pad. (a) Representative images of two single growing bacterial flagellum of cells immobilized on the agarose pad, taken at time points with 10-min time interval. Scale bar, 1 μm . **(b/c)** Growth measurements of 9 individual flagella tracked by real-time imaging. Plots of flagellar length versus time (b) and the flagellar growth rates versus flagellar lengths (c) are displayed. The flagella in (a) are emphasized by orange line with triangles (Top) and purple line with squares (Bottom). **(d)** Distribution of pause events (Blue) occurring in the total collected data points (Dark gray) at different flagellar lengths, binned at 200-nm intervals. All data points are from the measurements in (c).

5. The major new aspect of the injection-diffusion model developed by the authors is the implementation of fluctuations in the flagellin concentration. However, although the model can reproduce the experimental results, it remains unclear whether these fluctuations actually exist. Therefore, the value of the model remains unclear.

RESPONSE: To answer the reviewer's question about the actual state of intracellular flagellins, we performed time-lapse recording of cellular FliC-FIAsH fluorescence to validate the temporal fluctuation of flagellin proteins inside the cells. Briefly, the saturated culture of RP437-TC-FliC strain was diluted by 1:100 in M9 medium (M9 minimal supplemented with 0.4% glucose, 100x MEM vitamins, 50x amino acids and 10mM EDTA) and grown at 30°C to an appropriate concentration ($\text{OD}_{600}=0.3-0.4$). Labeling of intracellular flagellins was performed by gently mixing 5 μM FIAsH with the diluted *E. coli* cells and incubating for 30 min at 30°C. Time-lapse imaging was performed to record cell fluorescence with 10-min time intervals and ImageJ software was applied to select bacteria for fluorescence intensity quantification. For the majority of bacterial cells, newly born flagella occurred in the focal plane during the observation, which could not be used for analysis. In our results, 15 cells were

successfully measured and two typical examples of cell fluorescence recording were shown in the figure below. We observed that mean fluorescence intensity of cells exhibited variable changes with cell growth, indicating that flagellin density inside the cells underwent fluctuation.

Validation of the temporal fluctuation of intracellular flagellins. (a) Representative images of three fluorescent bacterial cells stained with FLaSH and recorded by 10-min interval time-lapse imaging. Scale bar, 1 μm . (b) Plots of mean fluorescence intensity (arbitrary unit) of single cells versus time.

Minor issues:

6. In the examples given in Figure 4a and b, the flagellar growth rates are below the population average. Is this true for all cases in which cells display multiple flagella? Are there also examples of cells carrying multiple flagella that all grow at rates higher than the population average? Are the growth rates measured in this case as similar as for the examples shown in Figure 4a and b?

RESPONSE: We thank the reviewer for raising this issue. In fact among the cells we analyzed, there are nearly no examples of cells carrying multiple flagella that grow at rates higher than the population average, except for some cells carrying only one flagellum. This suggests that the consumption of flagellins by multiple flagella on one cell is more likely to cause an insufficient flagellin supply and therefore the growth rates of multiple flagella on one cell would be below the population average.

7. In cells carrying the *fliC* overexpression plasmid (Figure 5b), the increase in the level of flagellin is surprisingly low, considering that the promoter system employed is routinely used for protein purification purposes. Do the authors have any explanation for that?

RESPONSE: The reason why the increase in the level of flagellin seems low is that the total amount of flagellin of bacterial cells was detected in Figure 5b, including intracellular flagellins and extracellular filaments. Considering that the assembled filaments outside the cell carry substantial flagellin proteins, the overexpressed flagellins only account for a very small fraction of the total flagellins and therefore do not result in a large increase in the blotting experiments.

To verify this, we removed flagella by shearing before performing western blotting and examined intracellular flagellins. As shown in the image below (relative flagellin levels report Mean \pm S.D., n = 3), for intracellular flagellin (two lanes on the right), the increase by overexpression is roughly 2-fold, which is apparently much higher than the total flagellins (two lanes on the left).

Despite removing assembled external flagella by shearing, there still remained some truncated external flagella which affected the total amount of flagellin measured using blots.

Validation of FliC overexpression. Immunoblotting of total flagellin (Left) and intracellular flagellin (Right) of RP437-TC-FliC strains with or without plasmid expression. The bands were quantitated by ImageJ (relative flagellin levels report mean \pm S.D., n = 3).

8. What is the total number of FliC molecules in a wild-type cell? Is this number really limiting, considering the number of molecules required to synthesize flagella at the rates measured in this study?

REPOSE: This is an interesting, important, and difficult question. Because flagellar growth is a dynamic process during cell growth, the total number of intracellular FliC molecules is the balance of FliC production and secretion. As one turn of 11 flagellins elongates the flagellum by 52 Å in *Salmonella enterica* (Yonekura et al., 2003), the growth increment of each folded FliC can be estimated as $52 \text{ Å} / 11 = 0.47 \text{ nm}$. From our measured *E. coli* flagellar growth, it requires ~50 flagellins per minutes per flagellum. If there are 3 flagella in a cell, it requires about 150 flagellins/min. To estimate the number of FliC molecules inside the cells, we measured the cell fluorescence and flagella fluorescence. By calculating mean fluorescence intensity of a single flagellin of the flagellum, the number of FliC inside the cells was obtained ranging from 300 to 700 flagellins. Therefore, the flagellin consumption is comparable to the intracellular FliC concentration. However, our overall knowledge of FliC production, secretion, and any feedback between these processes, is very limited such that we cannot easily calculate which step is the rate-limiting step. We

hope that our results raise this question for future investigations into this dynamic system.

9. The statement that bacteria live in environments of low Reynolds numbers refers to the motion of bacterial cells in liquid solutions. It is not clear to me why this should also apply to the motion of proteins encapsulated in the flagellar filament (line 327).

RESPONSE: We thank the reviewer for raising this concern. It is well accepted that bacteria in liquid solutions live at a low Reynolds number. Given the representative length scale of proteins and the viscous drag coefficient in liquid solution, previous research (Mogilner, A., H. Wang, T. Elston, G. Oster. *Molecular Motors: Theory and Experiment. Computational Cell Biology*. 2002.) has also shown that the motion of proteins is featured by a low Reynolds number and can be described by the Langevin equation which neglects the inertia term.

10. The authors state the rate at which new flagellin subunits are loaded by the export apparatus is proportional to the flagellin concentration. Do they assume a linear relationship? Shouldn't the rate depend on the affinity (K_d value) of the export apparatus for its substrate and thus show a non-linear dependency on the FliC concentration?

RESPONSE: We thank the reviewer for raising this question of modeling. We apologize for having been unclear in the manuscript. We did not simulate the binding process and the effect of FliC concentration on export. For simplicity, our simulation uses a varying loading rate (Eq. 9) to simulate the complete outcome of FliC secretion. We correct the description of the model in Line 340-342 to state clearly of our model.

Comments:

11. line 67: "with an optimized"
12. line 68: "with biarsenic dyes"
13. lines 96ff: "tetracysteine tag (...) -containing"
14. throughout the manuscript: "spinning-disc confocal microscopy"
15. line 126: "showed large variations"
16. throughout the paper: "fluorescence microscopy/imaging".
17. line 163: Why "in contrast"? This study leads to the same conclusion. It additionally suggests that there may be frequent pauses.
18. line 182: "growth of the"
19. lines 185-188: This part should be moved to the figure legend.
20. line 242: "expression of a plasmid-borne RP437-TC-fliC gene".
21. Supplementary Figure 4: Did both strains contain the overexpression plasmid or only the strain that was grown in the presence of arabinose?
22. line 257: "on flagellar growth"
23. line 263: "recovered by"

24. line 355: “follows”

25. line 395: “between”

26. References: Check for italicization of gene and species names, small print of titles, and proper abbreviation of journal names (e.g. references 7, 12, 15, 18, 20, 24, 25, 29, 31, 36).

RESPONSE: We greatly appreciate the comments of this reviewer and have made changes accordingly in our revised manuscript. For question 21, one strain is RP437-TC-FliC without plasmid and the other is RP437-TC-FliC containing the overexpression plasmid and we also clarified this in the legend of Supplementary Figure 6.

Reviewers' comments:

Reviewer #1 (Remarks to the Author):

[No further comments for author.]

Reviewer #2 (Remarks to the Author):

The authors have solved all raised questions in my opinion.

Reviewer #3 (Remarks to the Author):

In the revised version of the manuscript, the authors address most of the issues raised in the first round of review. Importantly, they include control experiments to show that the use of poly-L-lysine and the FIAsH/ReAsH labeling procedure have no adverse effects on the physiology of the cells or the growth dynamics of flagella. Moreover, they calculate approximate correlation coefficients for the growth rates of multiple flagella originating from the same cell and re-analyze the effect of FliC overproduction. The additional data make the paper much stronger, and there are only a few minor points that should be addressed before publication.

1. There are some issues with the new experiment investigating fluctuations in the level of FliC over time.

a) It is not perfectly clear to me how the authors performed this experiment. They state that the cells were labeled before the start of the timelapse series. However, in this case, only the FliC molecules present at the beginning of the experiment were labeled. Consequently, the fluorescence levels are expected to decrease as flagella grow but they should not be able to increase again at a later timepoint. Or was FIAsH present throughout the timelapse series to label all newly synthesized protein? If so, how did the authors account for the significant non-specific accumulation of the dye within the cells?

b) This experiment lacks a control that demonstrates that the changes in fluorescence intensities are not due to focal drift. The authors should include an analysis of fixed cells, which are expected to show constant fluorescence signals.

c) The authors state that the M9 medium used for this experiment contained 10 mM EDTA. Why is this? This concentration of EDTA may destabilize the LPS layer and thus affect cell growth. Can the authors exclude this possibility?

2. The Western blot testing FliC overexpression that is presented in the rebuttal letter should be included in the supplemental material. It is clearer than the one that is currently included in the main part of the paper.

3. I still think that the statement that bacteria live in an environment of low Reynolds number (as described in ref. 31) is not a good argument for the removal of the inertia term from the Langevin equation. After all, proteins do not care about the environment that their host lives in. It may well be that the assumption made by the authors is correct, but the reference given does not support their

claim.

Other comments:

- line 27: It is not the dye that is optimized and tagged.
- line 33: This sentence is somewhat unclear.
- line 85: "biarsenical dyes"
- lines 127: "E. coli showed"
- lines 416-419: This new section may not be easily comprehensible to the general reader. Please check for clarity and language.
- line 468: "the TC tag"

Reviewer 3

In the revised version of the manuscript, the authors address most of the issues raised in the first round of review. Importantly, they include control experiments to show that the use of poly-L-lysine and the FIASH/ReAsH labeling procedure have no adverse effects on the physiology of the cells or the growth dynamics of flagella. Moreover, they calculate approximate correlation coefficients for the growth rates of multiple flagella originating from the same cell and re-analyze the effect of FliC overproduction. The additional data make the paper much stronger, and there are only a few minor points that should be addressed before publication.

1. There are some issues with the new experiment investigating fluctuations in the level of FliC over time.

a) It is not perfectly clear to me how the authors performed this experiment. They state that the cells were labeled before the start of the timelapse series. However, in this case, only the FliC molecules present at the beginning of the experiment were labeled. Consequently, the fluorescence levels are expected to decrease as flagella grow but they should not be able to increase again at a later timepoint. Or was FIASH present throughout the timelapse series to label all newly synthesized protein? If so, how did the authors account for the significant non-specific accumulation of the dye within the cells?

RESPONSE: We thank the reviewer for raising this question. During the time-lapse recording, FIASH was always kept in the medium to ensure that all flagellins including newly synthesized ones can be labeled. Although it has been reported that FIASH dyes become strongly fluorescent only when they bind to the tetracysteine tag, there does exist some nonspecific background staining. To examine this nonspecific accumulation of the dye within the cells, we repeated the experiment with the RP437 wild-type strain that does not have the tetracysteine tagged flagellins. The non-specific fluorescence of 10 individual bacterial cells was measured and two representative examples of cellular fluorescence recording are shown in the figure below. We found that cellular autofluorescence of the RP437 wild type strain accounted for less than half of the cellular fluorescence of RP437-TC-FliC strain. Besides, the variation in fluorescence from non-specific binding in the RP437 wild type strain (~30A.U.) was far smaller than that in the RP437-TC-FliC strain (~400A.U.). These results indicate that non-specific dye accumulation inside the cells does not cause fluctuations as large as those observed in the RP437-TC-FliC strain (which were caused by the fluctuation in flagellin concentration). This confirmed our earlier conclusion that the flagellin density inside the cells undergoes dynamic changes.

Examination of non-specific accumulation of FIAsh dyes within the cells. (a) Representative time-lapse images of two RP437 wild type bacterial cells stained with FIAsh and recorded by a 10-min interval. Scale bar, 1 μm. **(b)** Plots of the mean fluorescence intensity (arbitrary unit) of single cells versus time.

b) This experiment lacks a control that demonstrates that the changes in fluorescence intensities are not due to focal drift. The authors should include an analysis of fixed cells, which are expected to show constant fluorescence signals.

RESPONSE: In the time-lapse recording of cellular FliC-FIAsh fluorescence, the Perfect Focus System (PFS) was always turned on to maintain focus in real time. This system automatically corrects focus drift caused by temperature changes and mechanical vibrations. We followed the reviewer's advice to add an analysis of fixed cells to rule out any effect from focal drift. Briefly, the saturated culture of RP437-TC-FliC strain was diluted by 1:100 in M9 medium (M9 minimal supplemented with 0.4% glucose, 100x MEM vitamins, 50x amino acids and 10mM EDTA) and grown at 30°C to an appropriate concentration ($OD_{600}=0.3-0.4$). Cells were then fixed with 4% PFA for 20 min and washed with M9 medium for two times. Labeling of intracellular flagellins was performed by gently mixing 5 μM FIAsh with the diluted *E. coli* cells and incubating the cells for 30 min at 30°C. Time-lapse imaging was performed to record cell fluorescence with 10-min time intervals and ImageJ software was used to select bacteria for fluorescence intensity quantification. In our results, 12 cells were successfully measured and a typical example of cell fluorescence recording is shown in the figure below. We observed that the mean fluorescence intensity of cells stayed constant and only reduced slightly towards the end of recording, indicating that the focus was maintained during observations over long times.

Validation of imaging focal plane with fixed cells. (a) Representative images of a fixed bacterial cell stained with FLaSH and recorded by time-lapse imaging at 10 min intervals. Scale bar, 1 μm. **(b)** Plots of mean fluorescence intensity (arbitrary unit) of single fixed cells versus time.

c) The authors state that the M9 medium used for this experiment contained 10 mM EDTA. Why is this? This concentration of EDTA may destabilize the LPS layer and thus affect cell growth. Can the authors exclude this possibility?

RESPONSE: We thank the reviewer for raising this question. Although the FLaSH dye is reported to be membrane-permeant (Ref 27 and 28), we found that in *E. coli* the dye cannot move into the cell very well. That's the reason why we used 10 mM EDTA to help FLaSH to enter the cell and ensure flagellins inside the cells are all labeled. To rule out the possibility that the addition of EDTA might affect bacteria growth, we checked the physiology of cells. Briefly, a saturated culture of the RP437-TC-FliC strain was diluted by 1:100 in M9 medium (M9 minimal supplemented with 0.4% glucose, 100x MEM vitamins, 50x amino acids and 10 mM EDTA). For a separate control group, no EDTA was added in M9 medium from beginning to the end. Both cultures were grown at 30°C to an appropriate concentration ($OD_{600}=0.3-0.4$). Labeling of intracellular flagellin was performed by gently mixing 5 μM FLaSH with the diluted *E. coli* cells and incubating for 30 min at 30°C. Time-lapse imaging was performed under bright-field microscopy. For each group, 169 cells were measured and the doubling time of the bacteria is shown in the figure below. No significant difference was observed between the two groups, indicating that 10 mM EDTA did not affect cell growth and viability.

Examination of the physiology of cells grown in M9 medium with 10mM EDTA. **(a)** Time-lapse bright-field images of bacterial growth recorded in M9 medium with (top) or without (bottom) 10mM EDTA. Scale bar, 1 μm . **(b)** Doubling time comparison between cells grown in M9 medium with (blue) or without (grey) 10mM EDTA. 169 cells for each group were analyzed. Mean \pm SD: Blue, 52.4 ± 14.2 min; Grey, 50.7 ± 12.6 min. Statistical analysis of two-tailed t-test was performed defining the difference as insignificant, $P = 0.24$.

2. The Western blot testing FliC overexpression that is presented in the rebuttal letter should be included in the supplemental material. It is clearer than the one that is currently included in the main part of the paper.

RESPONSE: We followed the reviewer's suggestion and included the validation of FliC overexpression by western blot in the Supplementary Information Line 133-146 and Supplementary Fig.8.

3. I still think that the statement that bacteria live in an environment of low Reynolds number (as described in ref. 31) is not a good argument for the removal of the inertia term from the Langevin equation. After all, proteins do not care about the environment that their host lives in. It may well be that the assumption made by the authors is correct, but the reference given does not support their claim.

RESPONSE: We thank the reviewer for pointing this out. We admit that our use of this reference may cause confusion. Given the size of a protein and its low viscous drag, proteins are also present in an environment of low Reynolds number. Ref 32 is a well-known publication from the literature studying protein motors and the motion of proteins. From page 332 to 333, it gives a proof of why the inertia term can be removed from the Langevin equation when describing the motion of proteins in a typical aqueous liquid environment. We have added this reference and clarified this in our revised manuscript (Line 329).

Other comments:

- line 27: It is not the dye that is optimized and tagged.
- line 33: This sentence is somewhat unclear.
- line 85: “biarsenical dyes”
- lines 127: “E. coli showed”
- lines 416-419: This new section may not be easily comprehensible to the general reader. Please check for clarity and language.
- line 468: “the TC tag”

RESPONSE: We appreciate the further help from the reviewer and we have made changes accordingly in our revised manuscript.

REVIEWERS' COMMENTS:

Reviewer #3 (Remarks to the Author):

The additional data fully address my concerns.